# Sequence-to-sequence translation from mass spectra to peptides with a transformer model

Melih Yilmaz [1,5], William E. Fondrie[2,5], Wout Bittremieux [3,5], Carlo F. Melendez[4], Rowan Nelson [4], Varun Ananth[1], Sewoong Oh[1] & William Stafford Noble [1,4] ✉

A fundamental challenge in mass spectrometry-based proteomics is the identification of the peptide that generated each acquired tandem mass spectrum. Approaches that leverage known peptide sequence databases cannot detect unexpected peptides and can be impractical or impossible to apply in some settings. Thus, the ability to assign peptide sequences to tandem mass spectra without prior information—de novo peptide sequencing—is valuable for tasks including antibody sequencing, immunopeptidomics, and metaproteomics. Although many methods have been developed to address this problem, it remains an outstanding challenge in part due to the difficulty of modeling the irregular data structure of tandem mass spectra. Here, we describe Casanovo, a machine learning model that uses a transformer neural network architecture to translate the sequence of peaks in a tandem mass spectrum into the sequence of amino acids that comprise the generating peptide. We train a Casanovo model from 30 million labeled spectra and demonstrate that the model outperforms several state-of-the-art methods on a cross-species benchmark dataset. We also develop a version of Casanovo that is fine-tuned for non-enzymatic peptides. Finally, we demonstrate that Casanovo's superior performance improves the analysis of immunopeptidomics and metaproteomics experiments and allows us to delve deeper into the dark proteome.

Mass spectrometry is currently the most popular analytical technique to characterize the proteome, by identifying and quantifying proteins present in complex biological systems[1]. During a bottom-up mass spectrometry proteomics experiment, proteins from a biological sample are enzymatically digested into peptides, their intact mass and charge are measured, and they are fragmented using tandem mass spectrometry. The fundamental challenge of mass spectrometry proteomics is then to determine the amino acid sequences of the resulting tandem mass (MS/MS) spectra. The standard approach to solve this spectrum identification problem is sequence database searching, during which peptides are simulated in silico using known digestion rules from a database of protein sequences potentially present in the biological samples, typically from a reference proteome. Next, each observed MS/MS spectrum is scored against a list of candidate peptides based on simplified peptide fragmentation rules, and the best-scoring peptide-spectrum match (PSM) is reported. Pioneered by the SEQUEST algorithm[2], dozens of database search engines have been subsequently developed and are very widely deployed[3].

[1]Paul G. Allen School of Computer Science and Engineering, University of Washington, Seattle, USA. [2]Talus Bioscience, Seattle, USA. [3]Department of Computer Science, University of Antwerp, Antwerp, Belgium. [4]Department of Genome Sciences, University of Washington, Seattle, USA. [5]These authors contributed equally: Melih Yilmaz, William E. Fondrie, Wout Bittremieux. ✉e-mail: William-noble@uw.edu

However, a fundamental requirement for sequence database searching is that the set of proteins that may be present in the sample is known in advance. While this is often the case for samples generated from species with well-characterized genomes, relying on a database prevents the detection of unexpected peptides. Such unexpected peptides include not just peptides derived from contaminant proteins or that arise due to variability in sample processing[4], but also biologically and clinically relevant peptides, such as peptides that deviate from the reference proteome due to genetic variation, peptides with unexpected post-translational modifications (PTMs), and peptides originating from foreign sources, such as microbes or consumed foods. Furthermore, there are tasks where generating a peptide database can prove impractical or even impossible. For example, the antigenic peptides presented by major histocompatibility complex (MHC) proteins—the "immunopeptidome"—are often generated from their parent proteins in an unpredictable manner, requiring at minimum every possible protein subsequence to be considered[5–7].

Similarly, constructing a peptide database for antibody sequencing is nearly impossible, due to the sequence variants created by V(D)J recombination[8]. Finally, creating an accurate database for mixtures of many organisms—metaproteomics—such as from microbiome or environmental samples, is often not feasible[9].

Such applications require the ability to sequence peptides directly from the acquired MS/MS spectra de novo. Early de novo peptide sequencing algorithms used heuristic search[10] or dynamic programming[11,12] to propose peptides for the observed MS/MS spectra. In addition to dynamic programming, the PepNovo algorithm[13] attempted to account for rules governing peptide fragmentation in its probabilistic score function and is closely related to the hidden Markov model employed by Fischer et al.[14] In 2015, the Novor algorithm[15] improved the state of the art by using a decision tree as the score function for its dynamic programming algorithm.

More recently, as in many other fields, deep learning has become the preferred solution for de novo peptide sequencing. DeepNovo[8] combines a convolutional neural network and a recurrent neural network to predict the subsequent amino acid when provided an MS/MS spectrum and a peptide prefix. SMSNet[16] uses a similar network architecture but reconciles the predicted sequences against a user-supplied peptide database. PointNovo[17], the successor to DeepNovo, leverages an order-invariant network architecture to model high-resolution MS/MS spectra[18]. Finally, pNovo 3[19] first generates candidate peptides for each MS/MS spectrum using dynamic programming, after which a final score is determined by matching the spectrum against a theoretical spectrum for each candidate peptide, simulated using the pDeep[20] learning-to-rank framework.

Despite the advances of these deep learning-based methods for de novo peptide sequencing, they still suffer from several limitations. De novo tools typically can only annotate a minority of MS/MS spectra compared to sequence database searching, they struggle with natively encoding high-resolution MS/MS data, and they employ complex neural network architectures and post-processing steps.

To address these issues, here we describe Casanovo, which reframes the de novo peptide sequencing task as a machine translation problem: like translating a sequence of words in a sentence from one language to another, Casanovo translates a sequence of peaks in an MS/MS spectrum into a sequence of amino acids of the generating peptide. To do so, we leverage the state-of-the-art architecture for modeling natural language—the transformer[21]. The transformer architecture allows Casanovo to directly use the $m/z$ and intensity value pairs that comprise an MS/MS spectrum without discretization of the $m/z$ axis and to directly output a predicted peptide sequence without a complicated dynamic programming step. We have previously[22] trained Casanovo on a limited collection of mass spectra from the multi-species benchmark used by Tran et al.[8] In this work, we present significant improvements to Casanovo and demonstrate its effectiveness

at tackling common challenges with de novo peptide sequencing. We expand our training set to use 30 million confident PSMs from the MassIVE-KB spectral library[23], and we add a beam search decoding procedure to predict the best peptide for each MS/MS spectrum. We demonstrate that, together, these updates significantly improve Casanovo's already state-of-the-art performance. Additionally, we fine-tune a non-enzymatic version of Casanovo for tasks such as immunopeptidomics. We demonstrate how high-performance de novo peptide sequencing using Casanovo enables fast and effective immunopeptidome analysis, bolsters the characterization of metaproteomes, and sheds light on the dark proteome.

## Results

### A transformer architecture enables processing of raw mass spectra

Casanovo uses a transformer architecture to perform a sequence-to-sequence modeling task, from MS/MS spectrum to the generating peptide (Fig. 1). Transformers are built upon the attention function[21], which allows transformer models to contextualize the elements of a sequence; transformer models thus learn the relationships of sequence elements to one another and how their interactions should be interpreted. As such, the transformer architecture has found success in not only natural language processing, but also applications to biological sequences[24,25].

In Casanovo, each peak in an observed MS/MS spectrum is considered as an element in a variable length sequence. The $m/z$ and intensity values of each peak are encoded using, respectively, a collection of sinusoidal functions and a learned linear layer, and these encodings are summed. The encoded peaks are then input into the transformer encoder, where context is learned between pairs of peaks in the MS/MS spectrum. The contextualized peak encodings are then used as input to the transformer decoder for predicting the peptide sequence.

The process of decoding proceeds in an iterative, autoregressive manner. We begin by providing the mass and charge of the observed precursor. The transformer decoder uses the contextualized peak encodings and the precursor information to begin predicting amino acids of the peptide. With the first predicted amino acid, we retain the top $k$ residues, where $k$ is a user-selected value for the number of

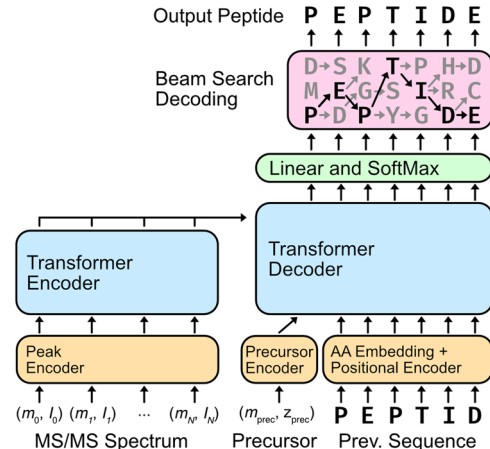

**Fig. 1 | Casanovo performs de novo peptide sequencing using a transformer architecture.** The peaks from each MS/MS spectrum are contextualized by the transformer encoder. The resulting peak encodings are then fed into the transformer decoder along with the observed precursor mass and charge to iteratively decode the peptide sequence. Casanovo uses a beam search decoding strategy, following the most promising sequence predictions until they terminate or exceed the precursor mass. The highest scoring peptide sequence is returned as the putative peptide that generated the MS/MS spectrum.

beams in our beam search. In each subsequent iteration, amino acids are added to the decoded peptide sequence, retaining the top $k$ sequences until the decoded sequences for all of the beams have terminated or exceeded the precursor mass. Finally, the sequence with the highest score is retained as the putative peptide that generated the provided MS/MS spectrum.

In generating its predictions, Casanovo will inevitably fail to generate plausible peptides for some MS/MS spectra. For example, some MS/MS spectra contain too few fragment ions to be sequenced reliably, or the true generating peptide may bear a modification that is unknown to Casanovo. We therefore refine the PSMs proposed by Casanovo using a simple precursor mass filter: any PSMs for which the $m/z$ of the peptide falls outside the specified tolerance of the observed precursor, including potential isotopes, is discarded. This filter eliminates many poorly scored peptides from consideration. In our evaluations, PSMs that would normally be removed by this filter were retained and ranked last among all PSMs assigned by Casanovo.

### Casanovo outperforms state-of-the-art methods

To evaluate Casanovo, we first used the nine-species benchmark dataset originally created by Tran et al.[8] to compare the performance of four de novo peptide sequencing algorithms: Novor, DeepNovo, PointNovo, and Casanovo. For these comparisons, we used the publicly available, pretrained version of Novor to sequence the MS/MS spectra in the benchmark dataset. DeepNovo, PointNovo and Casanovo were trained in a cross-validated fashion, systematically training on eight species and testing on the remaining species. For DeepNovo, we used the models trained and provided by Tran et al.[8] for each of the cross-validation splits. For PointNovo, we cross-validated nine models from scratch using the code and settings provided by Qiao et al.[17]. This benchmark version of Casanovo, Casanovo$_{bm}$, employs a simple greedy decoding algorithm, rather than beam-search decoding. The results (Fig. 2a) revealed that Casanovo$_{bm}$ substantially improved peptide-level sequencing performance over Novor, DeepNovo and PointNovo, with an average precision of 0.81 for Casaonovo$_{bm}$ compared to 0.58, 0.70 and 0.74 for Novor, DeepNovo and PointNovo, respectively. These results are consistent across all nine species in the benchmark dataset (Supplementary Fig. S1).

We hypothesized that Casanovo could achieve even better performance if provided with a larger training set of higher quality PSMs; hence, we turned to the MassIVE-KB spectral library of human MS/MS

proteomics data[23]. MassIVE-KB provided us with a set of 30 million high confidence PSMs, which we previously collected for training our GLEAMS embedding model[26]. This dataset contains not only a greater diversity of peptides and MS/MS spectra generated from multiple instruments, but also additional types of post-translational modifications. We therefore created a new version of the nine-species benchmark dataset using the same nine PRIDE datasets but including seven different types of variable modifications (methionine oxidation, asparagine deamidation, glutamine deamidation, N-terminal acetylation, N-terminal carbamylation, N-terminal NH$_3$ loss, and the combination of N-terminal carbamylation and NH$_3$ loss). In the process, we also fixed several problems that we uncovered in the previous benchmark, including adding consideration of isotope errors and eliminating peptides that occur in multiple species (see Methods for details). The final, revised benchmark dataset consists of 2.8 million PSMs drawn from 343 RAW files.

The results from evaluating with respect to this revised benchmark demonstrate the value of training from a much larger collection of higher quality PSMs (Fig. 2b). When trained on the MassIVE-KB dataset, the average precision of Casanovo increases from 0.83 to 0.95. Furthermore, Casanovo succeeds in making a larger proportion of predictions with $m/z$ values that fall within 30 ppm of the observed precursor (signified by the location of the diamonds in Fig. 2b), increasing from 70% to 97%. Additionally, an analysis of spectrum identifications for all de novo sequencing tools on the nine-species benchmark dataset shows that correct Casanovo PSMs include almost all correct identifications of the competing de novo sequencing methods, as well as approximately 50% more correct PSMs that are unique to Casanovo (Supplementary Fig. S8). This version of Casanovo incorporates beam-search decoding, which improves both average precision and coverage compared to greedy decoding for the same model (Supplementary Fig. S3).

In one sense, this comparison is unfair because some of the spectra in the new version of the benchmark contain PTMs that cannot be identified by some of the competing methods. We therefore eliminated these spectra from each test set and then re-computed the precision-coverage curve. The results (Supplementary Fig. S4) are largely unchanged, suggesting that the PTMs contribute little to the observed overall differences in performance.

To better understand why the model trained on MassIVE-KB outperforms the one trained on the nine-species benchmark, we

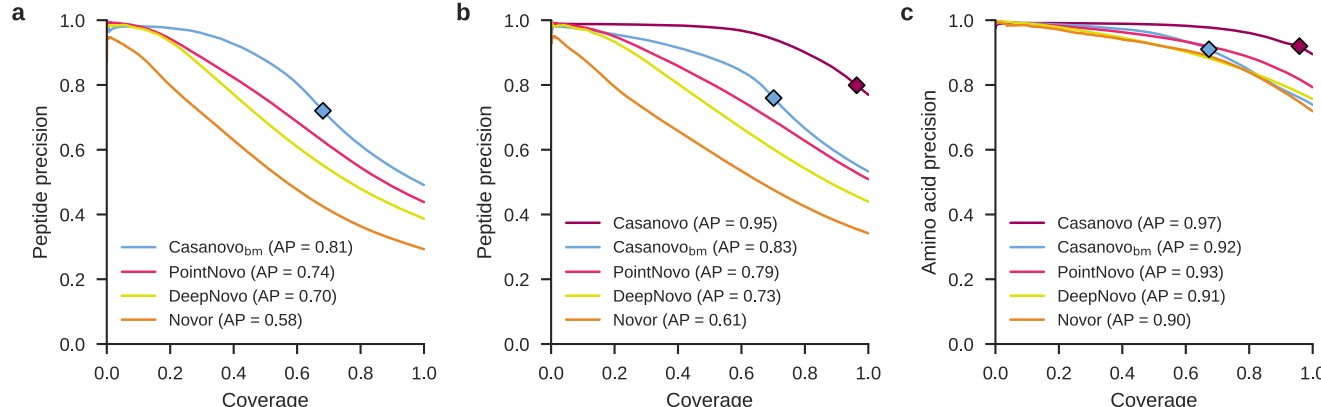

**Fig. 2 | Casanovo outperforms PointNovo, DeepNovo, and Novor on a nine-species benchmark. a** Casanovo maintains high peptide-level precision (the proportion of correctly predicted peptides) across all values of coverage (the proportion of spectra for which a prediction is made). Each curve is computed by sorting predicted peptides for all nine species according to their peptide-level confidence scores. For Casanovo, all peptides that pass the precursor $m/z$ filter are ranked above peptides that do not pass the filter, and the boundary is indicated by a diamond on each curve. Average precision (AP) corresponds to the area under the precision-coverage curve. **b** Same as **a**, but using the revised benchmark and including a version of Casanovo trained on MassIVE-KB. **c** Casanovo's amino acid-level precision is greatly improved by the expanded training data provided by MassIVE-KB. The test set is the revised nine-species benchmark, with PSMs only containing modifications considered by both DeepNovo and Casanovo.

performed two follow-up experiments. First, we trained a series of Casanovo models on randomly sampled nested subsets of MassIVE-KB, ranging from 250,000 spectra to the full dataset of 28 million spectra. Each model was then evaluated with respect to the revised nine-species benchmark. The resulting learning curve (Supplementary Fig. S5) shows that the test set performance depends strongly on the size of the training set, though with diminishing returns after a million or so PSMs. Second, we directly compared a Casanovo model trained from a downsampled MassIVE-KB dataset to Casanovo$_{bm}$ which averages 9 models cross-validated on the nine-species benchmark, where the training sets contain approximately the same number of peptides (239,697 for Massive-KB and 246,713 for Casanovo$_{bm}$). We then evaluated both models using the revised nine-species benchmark. The results (Supplementary Fig. S6) show that the model trained from MassIVE-KB substantially outperforms Casanovo$_{bm}$, with the average precision increasing from 0.83 to 0.90. Thus, these results suggest that the improved performance of the MassIVE-KB model stems primarily from the improved quality of the data rather than the size of the data set.

In addition to evaluating Casanovo's ability to correctly predict whole peptides, we also evaluated Casanovo's ability to predict the individual amino acids of each peptide. We did so by ranking amino acids by their associated confidence score and then plotting a precision-coverage curve. We compared two versions of Casanovo (trained from the first benchmark and from MassIVE-KB) with Deep-Novo and PointNovo on the revised nine-species benchmark with new modifications eliminated (Fig. 2c). The amino acid-level performance was consistent with the trends we observed in peptide-level performance, with Casanovo outperforming Novor, DeepNovo and Point-Novo: Casanovo trained on MassIVE-KB achieves a remarkable average precision of 0.98.

Finally, to characterize the improved de novo sequencing performance of Casanovo across generating peptides of different lengths and precursor charge states, we compare all methods on subsets of the revised nine-species benchmark dataset. First, we divide spectra into three groups by charge state where groups contain precursors with 2+, 3+ and 4+ or higher charge states each, and plot peptide precision-coverage curves for each group (Supplementary Fig. S9). As expected, average precision is lower across all methods for groups with higher precursor charge states since those spectra tend to have more complex fragmentation patterns. However, the drop in performance is only 12% for Casanovo in precursors with 4+ or higher charge states versus precursors with 2+ charge states, thanks to the diversity of precursor charge states in its training data where 11% of precursors have 4+ or higher, whereas average precision for all competing methods decreases by more than 60%. Second, we bin spectra according to the length of their generating peptides into groups of short (fewer than 13 amino acids), medium (between 13 and 18 amino acids), and long (greater than 18 amino acids) peptides, and compare de novo sequencing performance in each group (Supplementary Fig. S10). Performance degrades for longer peptides because incorrect amino acid predictions tend to accumulate during decoding, but the observed decrease in average precision for Casanovo is much smaller relative to other methods, highlighting Casanovo's ability to accurately sequence long peptides as a key contributor to its improved performance.

## Casanovo unravels the immunopeptidome

One important application of de novo peptide sequencing is the characterization of the peptides presented by major histocompatibility complexes (MHCs), which is commonly referred to as the "immunopeptidome." These antigen peptides are presented on the extracellular surface and serve as targets for immune cell recognition. However, because these antigen peptides are generated through lysosomal or proteasomal degradation, they do not exhibit the characteristic tryptic termini from most proteomics experiments.

Consequently, the peptide search space is exponentially larger than considering only tryptic peptides—every peptide subsequence in a protein within a defined peptide length must be considered. Furthermore, mutations in these peptides are of particular interest, because these mutation-containing neoantigens may serve as tumor-specific markers to activate T cells and initiate antitumor immune responses. Unfortunately, expanding the search space to consider all possible mutated peptides is prohibitive both in terms of search speed and statistical power for traditional proteomics search engines.

Although immunopeptidomics is a prime application for de novo sequencing, naively applying Casanovo directly to immunopeptidomics data is problematic: the standard Casanovo model is heavily biased to predict tryptic peptides due to their overrepresentation in MassIVE-KB. To demonstrate this effect, we analyzed five mass spectrometry runs generated from MHC class I peptides isolated from MDA-MB-231 breast cancer cells[5] in two different ways: first using Casanovo and second by searching against a non-enzymatic digestion of the human proteome (see Methods). Among the peptides accepted at 1% FDR by the database search procedure, we observed a low proportion of "tryptic" peptides, i.e., peptides with C-terminal amino acids of K (1.12%) or R (0.80%). In contrast, among the top-scoring 10% of the Casanovo predictions, we observed a greater than six-fold increase in the rate of tryptic peptide predictions (5.87% K and 6.76% R).

We hypothesized that we could reduce this tryptic bias and produce a version of Casanovo that is better suited to immunopeptidomics data by fine tuning our existing model using data that lacks a tryptic bias. To create such a dataset, we combined data from two sources. First, we segregated PSMs from MassIVE-KB according to their C-terminal amino acid and then uniformly sampled up to 50,000 peptides within each group. For most amino acids, MassIVE-KB contained fewer than 50,000 PSMs, so for these we supplemented by randomly extracting additional PSMs from the PROSPECT collection[27] (Supplementary Table S1). We then split this new collection of 1 million PSMs into training, validation, and testing sets. We then fine-tuned our existing Casanovo model by training it until convergence on this non-enzymatic training set.

The resulting model, Casanovo$_{ne}$, performs markedly better than the original Casanovo model at predicting peptides in our held-out, non-enzymatic test set. On the held-out test set of 100,000 non-enzymatic PSMs, Casanovo$_{ne}$ achieves an average precision of 0.83, compared with 0.60 for the original Casanovo model on the same data (Fig. 3a). The predicted C-terminal amino acid frequencies are also much closer to the true frequencies, with K and R dropping to 1.81% and 1.79%, respectively (Fig. 3b).

We next used Casanovo$_{ne}$ to sequence the immunopeptidome of MDA-MB-231 breast cancer cells[5]. For each peptide predicted by Casanovo, we investigated whether it (1) occurs anywhere within the human proteome, and (2) occurs within the set of peptides detected using a database search procedure. We first searched the data against a non-enzymatic digestion of the human proteome using the Tide search engine[28] followed by Percolator post-processing[29], using settings similar to those in the original study[5] (see Methods). Out of 26,377 unique peptides predicted by Casanovo, 2459 match to the human proteome, and a majority of these overlap with the 1544 unique peptides identified by Tide at 1% FDR (Supplementary Fig. S11). Notably, these overlapping peptides are predicted with high confidence by Casanovo, almost all within the first 10,000 Casanovo predictions (Fig. 3c). Casanovo predicts an additional 1148 peptides that match to the human proteome but are not identified by Tide at 1% FDR, and further analysis shows that 751 (65.4%) of these peptides correspond to Tide hits that were not accepted at the 1% FDR threshold. To further investigate the plausibility of Casanovo predicted peptides as MHC antigens, we used NetMHCpan-4.1[30] to predict MHC binding affinity for these peptides. First, we

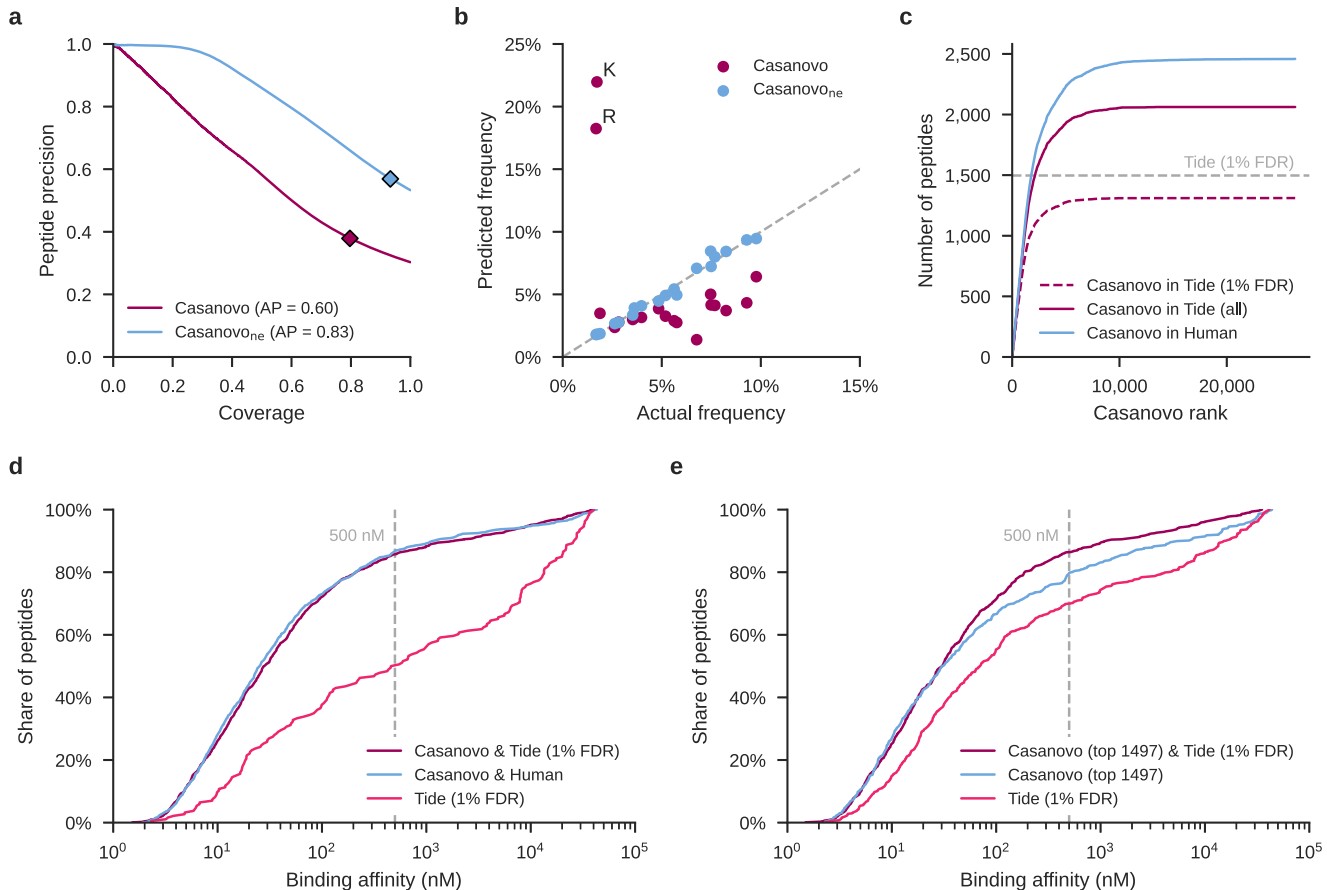

**Fig. 3 | Fine-tuning reduces Casanovo's bias for tryptic peptides. a** Fine-tuning Casanovo (Casanovo_ne) improves peptide-level precision on sequencing MS/MS spectra generated by non-tryptic peptides. **b** Casanovo_ne predicts non-tryptic C-terminal peptides more readily than the standard Casanovo model, improving performance on the non-enzymatic validation set. **c** Casanovo detects many peptides that are present in the human proteome but are not detected via database search. The dashed dark pink line only includes peptides detected by database search within the 1% FDR threshold, whereas the solid dark pink line includes all peptides from the database search, irrespective of FDR threshold. **d** The peptides proposed by Casanovo generally have higher predicted binding affinities for the MHC class I receptor, matching the performance of a Tide database search. The vertical bar corresponds to the 500 nM binding affinity below which peptides are predicted to be MHC binders. **e** Similar to **d**, but considering only the 1497 peptides that are accepted at 1% FDR by Tide which yielded valid binding affinity predictions from NetMHCpan and a corresponding set of 1497 highest-confidence Casanovo peptides.

compared peptides that were identified by both Casanovo and database search with peptides that were predicted only by Casanovo and match to the human proteome. These two groups exhibit similar distributions of predicted binding affinity profiles, with 87% of peptides identified by Casanovo alone and 86% of those identified by both methods predicted to be MHC binders at 500 nM (Fig. 3d). In contrast, when we evaluate peptides that are identified by database search but not by Casanovo, the proportion of predicted MHC binders drops substantially to 50%. Overall, these results suggest that Casanovo not only identifies more peptides matching to the human proteome than the standard database search procedure, but the peptides Casanovo predicts are also more likely to bind MHC antigens.

We also explored an alternative method for comparing Casanovo and Tide results, which does not rely on mapping Casanovo predictions to the reference proteome. For this analysis, we consider the 1497 peptides identified by Tide at 1% FDR which yielded valid binding affinity predictions from NetMHCpan alongside the top 1497 highest confidence Casanovo predictions. The results (Fig. 3e) agree with those in Fig. 3d: the 960 peptides in common between the two sets of peptides achieve the highest proportion of MHC binders (86%), the Casanovo-only predictions achieve a slightly lower percentage (80%), and the Tide-only predictions have the lowest percentage of MHC binders (70%).

## Casanovo accurately sequences peptides from complex metaproteomes

Proteomics applications extend far beyond the analysis of single model organisms or well-characterized biological systems. Indeed, there is growing interest in using mass spectrometry proteomics methods to investigate the dynamics of complex biological ecosystems—whether microbiomes or environmental specimens—for which the identities of its members cannot be known a priori. Due to the unknown complexity of the sample and even the lack of reliable reference proteomes for the likely species in the sample, these metaproteomics experiments are difficult to analyze. One solution to these problems is to search the spectra against a large database, such as one containing all the microbial sequences in public databases for a sample that is likely dominated by unknown microbes. This "big database" approach is widely used but suffers from a significant loss in statistical power due to the implicit multiple hypothesis testing correction that must be made to account for the size of the database. An alternative solution involves first subjecting the sample to genome sequencing, and then using the inferred peptide sequences as the basis for a "metapeptide" database. This approach yields better power to detect peptides[31] but requires the availability of a matched DNA sample and the additional cost associated with DNA sequencing.

We hypothesized that Casanovo's improved de novo sequencing capabilities would be useful in both scenarios—either in the presence

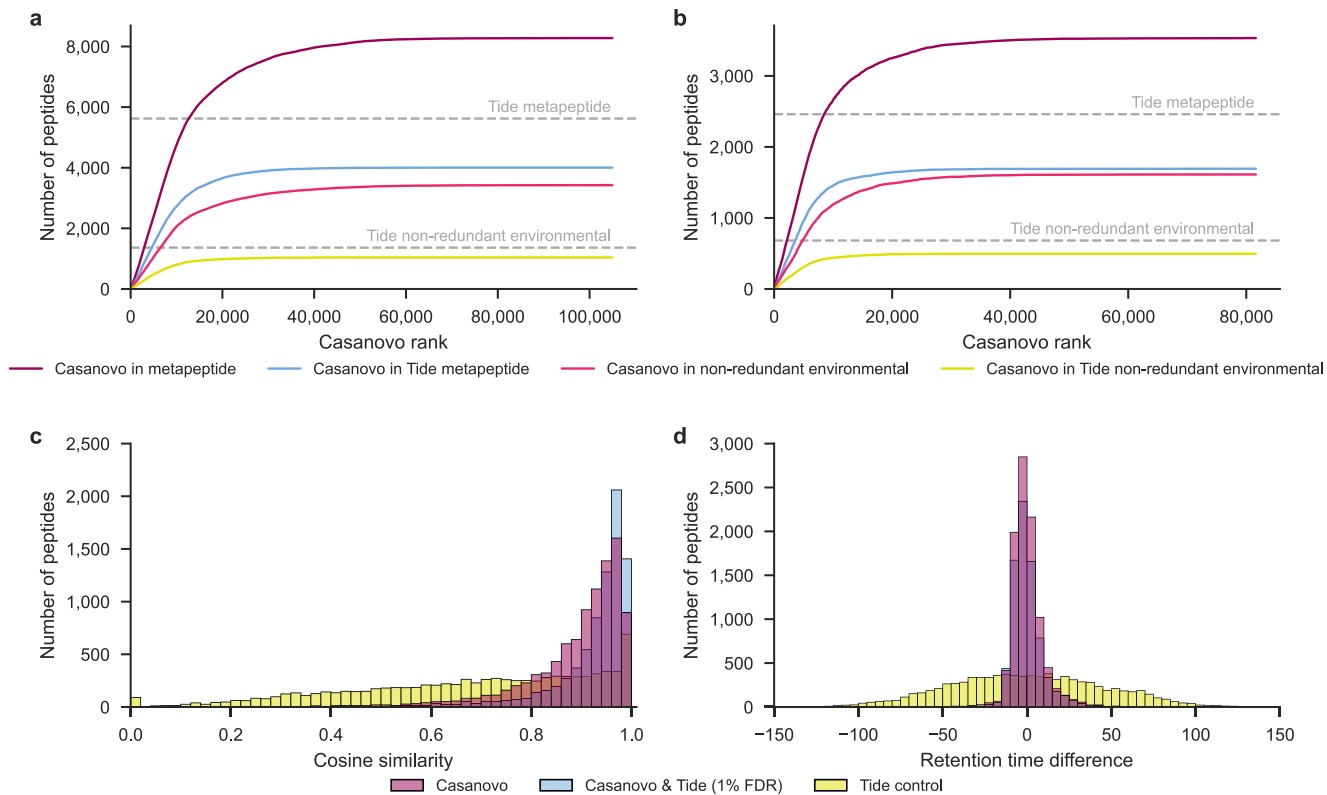

**Fig. 4 | Casanovo improves power to detect peptides from metaproteomics samples.** Casanovo assigns more peptides matching the metapeptide database and the non-redundant environment database than Tide and Percolator at 1% FDR in seawater samples from **a** the Bering Sea and **b** the Chukchi Sea. Peptides are ranked according to the Casanovo confidence score, assigning each peptide the maximum score across all three runs from each sampling location. Horizontal lines indicate the total number of distinct peptides detected by Tide+Percolator, searching against two different databases. **c** The PSMs assigned by Casanovo at a 1% error rate and Tide and Percolator at 1% FDR have high cosine similarities to the predicted MS/MS spectra for the respective peptides from Prosit when compared to control PSMs sampled from the Tide search results with > 10% FDR. Each group represents the aggregated results for Bering and Chukchi Sea data, as well as non-redundant environmental and metapeptide databases. **d** The PSMs assigned by Casanovo at a 1% error rate and Tide and Percolator at 1% FDR closely align with the predicted retention times from Prosit.

or absence of a metapeptide database. To test this hypothesis, we applied Casanovo to data from six previously published ocean meta-proteomics samples, three from the Bering Strait and three from the Chukchi Sea[31]. Critically, these samples were also subjected to DNA sequencing; hence, in addition to the non-redundant environmental database, we also have a metapeptide database for each sampling location.

We began by measuring the extent to which peptides detected by Casanovo occur within the corresponding metapeptide database or within the larger, non-redundant protein database. Because these samples were digested using trypsin, we used the standard Casanovo model, trained from the tryptic MassIVE-KB dataset. To control the error rate for the matching of Casanovo predictions to these data-bases, we employed a procedure similar to target-decoy competition used in the false discovery rate estimation for database search (see Methods for details) and only considered as correct Casanovo pep-tides found in the corresponding database that fall within the 1% ran-dom matching threshold. Using this logic, we obtain much better power to detect peptides using Casanovo than using a standard database search procedure against the metapeptide database (Fig. 4a–b). In particular, when we search the data against the meta-peptide database using Tide followed by Percolator[29], we detect 5623 peptides at 1% FDR in the Bering Strait data and 2460 peptides in the Chukchi Sea data. In contrast, if we run Casanovo and accept as correct only peptides that appear in the metapeptide database (subject to our 1% random matching criterion), then we detect 8277 and 3532 pep-tides, respectively, in the two datasets, representing increases of 47%

and 44%. Casanovo also outperforms database search when we con-sider the non-redundant protein database rather than the sample-specific metapeptide database. We detect 1364 peptides in the Bering Strait data and 682 peptides in the Chukchi Sea data at 1% FDR by searching the non-redundant environmental database using Tide and Percolator. In comparison, Casanovo predictions, filtered at 1% error rate using the environmental database, detect 3425 and 1612 peptides, respectively, representing increases of 151% and 136%, respectively.

When using both metapeptide databases or the non-redundant environment database, Casanovo detects most of the peptides iden-tified by Tide database search and Percolator, where it respectively detects 71% and 75% of Tide identifications on metapeptide and non-redundant databases, while also detecting a substantial number of additional unique peptides (Supplementary Fig. S12).

To validate the peptides that were detected by Casanovo but not the database search, we used the Prosit machine learning tool[32] to predict spectrum peak intensities and retention times for peptide identifications. First, we compared the cosine similarities between the observed and predicted MS/MS spectrum peak intensities across three groups of peptides: peptides only predicted by Casanovo that mat-ched to the database with 1% error, peptides detected both by Casa-novo and by Tide and Percolator at 1% FDR, and a control group of peptides detected by Tide and Percolator with >10% FDR. The control group was randomly sampled to be the same size as the Casanovo-only group. The results (Fig. 4c) indicate that the Casanovo-only identifi-cations have a high concentration of high cosine similarity peptides, similar to the overlapping identifications between Casanovo and

database search. This stands in contrast with the control group, which exhibits a much broader distribution of cosine similarities.

Second, we compared the observed retention times with the predicted retention times from Prosit for the same three groups of peptides. For each group, we calibrated the predicted retentions times to the observed retention times using linear regression (Supplementary Fig. S13). We observed that the peptides detected only by Casanovo and those detected by Casanovo and Tide had a similar slope and resulted in similar residual distributions (Fig. 4d). When compared against the control group, the residual distributions for peptides only detected by Casanovo and those detected by Casanovo and Tide are close to zero.

Ultimately, Casanovo does not yet allow us to achieve as much power with the non-redundant database as with the metapeptide database. For example, for the Bering Strait data, the union of the 3425 peptides detected using Casanovo and the 1364 peptides detected using database search is 3750, which is fewer than the 5623 peptides detected using the metapeptide database. (The corresponding numbers for the Chukchi Sea data are 1798 and 2460.) This difference is perhaps not surprising, because the environmental non-redundant database is incomplete: 3715 of the 5623 peptides found by the database search procedure in the Bering Strait metapeptide database are not even present in the environmental database. Thus, a rigorous FDR control procedure for de novo peptide sequencing is needed in order to rescue the many peptides that are correctly detected by Casanovo but cannot be validated by matching to a database.

## Casanovo shines a light on the dark proteome

The "dark matter" of mass spectrometry-based proteomics consists of MS/MS spectra that are observed repeatedly across experiments but consistently fail to be identified. In many cases, these MS/MS spectra may have been generated by peptides that are not in the canonical human proteome, because they represent contaminant peptides, result from non-standard enzymatic cleavage, or contain sequence variants. We hypothesized that Casanovo could shed light on some of this dark matter.

Accordingly, we applied Casanovo to a collection of MS/MS spectra drawn from a previous analysis[26], in which 511 million human spectra from MassIVE were grouped into 60 million clusters, and the clusters were systematically analyzed using targeted open modification searching of representative spectra. The analysis yielded a collection of 39 million unidentified clusters, containing a total of 207 million MS/MS spectra. For our analysis, we selected 3.4 million of these unidentified, clustered MS/MS spectra from eight randomly selected MassIVE datasets. These MS/MS spectra belong to 573,597 distinct clusters. Because we were investigating spectra that had already failed to be identified using a standard, tryptic pipeline, we opted to use the non-enzymatic Casanovo model (Casanovo$_{ne}$) to assign a peptide to each selected MS/MS spectrum, eliminating peptides for which the predicted m/z falls outside the associated mass range. This analysis yielded a total of 1.3 million predicted peptides.

We sought to ascertain how well Casanovo had assigned peptide sequences to these dark matter clusters by addressing this question in two complementary ways. First, we identified all clusters in which a plurality (and at least two) of the spectra were assigned to the same peptide sequence, and then we mapped those peptides to the human reference proteome, allowing at most one amino acid mismatch. The first step of this procedure assigns peptides to 89,250 (16%) of the clusters, of which 65% could be matched to the human proteome. The clusters identified in this fashion vary in size, ranging from 2 to 542 spectra per cluster, but when we limited the above analysis only to clusters larger than a certain size, we observed that the shares of identified clusters more than doubled (Supplementary Fig. S14). Second, we performed a complementary analysis, first eliminating all predicted peptides that do not occur within the human proteome

(again, allowing one mismatch) and then finding clusters with two or more spectra assigned the same sequence and no other spectrum assigned to a different sequence. This procedure assigns peptides to 52,523 clusters, corresponding to 9% of all previously unidentified clusters. The overlap between the two approaches—plurality vote followed by proteome matching or vice versa—is high: 98% of the 52,523 clusters overlapped with the clusters from the previous analysis. Overall, Casanovo is able to assign peptides to 196,724 of the 3.4 million unidentified MS/MS spectra using the combination of these two strategies.

One potential reason for an MS/MS spectrum to remain unidentified is the presence in the generating peptide sequence of a genetic variant that does not appear in the reference proteome. To investigate whether Casanovo is identifying such sequences, we looked more closely at the subset of Casanovo cluster assignments that match to the human proteome with a single amino acid mismatch, focusing on the 51,555 assignments that agree between the two methods described above. Two pieces of evidence suggests that these peptides are indeed enriched for genetic variants. First, we observe an enrichment for amino acid substitutions that can be explained by a corresponding single-nucleotide substitution. Among the Casanovo predictions, 83.4% correspond to a potential single-nucleotide substitution, compared with only 38.6% of all possible amino acid substitutions that fit this criterion. Second, we see a strong enrichment for substitutions with positive BLOSUM62 scores[33]. The BLOSUM score is an integerized log-odds score indicating the empirical substitutability of one amino acid for another. In the BLOSUM62 matrix, only 11% of the 380 non-diagonal entries are positive. However, if we rank the Casanovo-predicted substitutions by frequency, we find that the top five substitutions have BLOSUM scores of 1 or 2 (Supplementary Table S3). This observation strongly suggests that Casanovo is predicting substitutions that are biochemically plausible.

## Discussion

Casanovo's excellent performance derives from two sources: the availability of a large, high-quality set of training data, and the use of a machine learning architecture that can make use of that data. Our experiments suggest that the carefully curated MassIVE-KB collection provides particularly good training data. This is likely because the dataset was derived from a massive collection of 669 million spectra, in combination with extremely stringent FDR control. In particular, the data were searched at 1 % FDR, after which only the top 100 PSMs for each unique precursor were retained, corresponding to 30 million high-quality PSMs (uniformly 0 % FDR from the original searches).

The transformer architecture is uniquely suited to contextualize the elements of variable length sequences and has therefore proven immensely successful in modeling natural language. In comparison to recurrent neural networks, the transformer architecture is able to learn long-range dependencies between the elements of a sequence and can be parallelized for efficient training. By encoding the peaks of a mass spectrum as a sequence, similar to tokenizing the words of a sentence, Casanovo leverages the strengths of the transformer architecture and the rapid advances pioneered in large language models to improve de novo peptide sequencing from MS/MS spectra[34]. One important open question, which we leave for future work, is how the number of model parameters impacts de novo sequencing performance.

Casanovo's utility extends beyond the applications we have demonstrated here. Most obviously, any application in which a peptide database is unavailable, incomplete, or extremely large may benefit from de novo sequencing, such as paleoproteomics, forensics, or astrobiology[35]. However, even in the analysis of human or model organism data, Casanovo can assist in the detection of "foreign" spectra, i.e., spectra generated by peptides that are not present in the database. Such foreign spectra might correspond to contaminants introduced during the experiment itself, but they can also represent

microbial species, genetic variation, or trans-spliced peptides. In general, we envision applying Casanovo as a post-processor for spectra that fail to be assigned a peptide during a standard database search procedure, akin to the last stage of a cascade search[36].

One important application of de novo sequencing that we have not yet explored is antibody sequencing. However, a recent publication carried out a systematic comparison of six de novo sequencing tools, including Casanovo, on the problem of antibody sequencing[37]. The results show that Casanovo strongly outperforms the competing methods by all of the measures that the authors considered. Notably, this comparison employed a version of Casanovo that used greedy decoding and was trained on only 2 million spectra. Hence, our results (Fig. 2b) suggest that the version of Casanovo trained from 30 million spectra will yield even better antibody sequencing performance.

We anticipate many opportunities for fine-tuning the Casanovo model for particular applications. Our analysis with the non-enzymatic model suggests that Casanovo's enzymatic bias can be adjusted by using a relatively small amount of training data. Thus, in the short term, we plan to train variants of Casanovo that are appropriate for a variety of different cleavage enzymes. The Casanovo software makes such fine-tuning straightforward, so any user interested in adapting the model to a particular experimental setting should be able to do so. Longer term, an ideal model would take as input a spectrum along with relevant metadata, such as the digestion enzyme, collision energy, and instrument type, and predict accurately for many different types of experimental settings.

The potential for deep learning methods to improve our ability to perform de novo sequencing has now been widely recognized. While this paper was under review, at least six additional deep learning de novo sequencing methods have been published, including GraphNovo[38], PepNet[39], Denovo-GCN[40], Spectralis[41], π-HelixNovo[42], and NovoB[43]. Clearly, the field would benefit from an exhaustive and rigorous benchmark comparison of this growing field of tools.

On a related note, at this stage one of the key bottlenecks in the field is the absence of a rigorous method for confidence estimation for de novo sequencing. In our metaproteomics analysis, we have matched Casanovo predictions to a target and corresponding decoy peptide database, but such an approach misses out on the power of de novo sequencing to assign peptides to foreign spectra. Thus, an open question is whether, for a given data-dependent acquisition dataset, Casanovo outperforms a standard database search procedure in terms of statistical power to detect peptides. Trained from sufficiently large training sets, we may be approaching the end of database searching as the go-to method for analysis of DDA tandem mass spectrometry data.

## Methods

### Casanovo

Casanovo consists of a transformer encoder and decoder stack as described by Vaswani et al.[21], which are respectively responsible for learning latent representations of the input spectrum peaks and decoding the amino acid sequence of the spectrum's generating peptide. The encoder takes $d$-dimensional spectrum peak embeddings as input and outputs $d$-dimensional latent representation vectors for each peak. Subsequently, the decoder takes as input these representations of prefix amino acids, coupled with a $d$-dimensional precursor embedding encapsulating precursor $m/z$ and charge information, to predict the next amino acid in the peptide sequence. We discuss different aspects of our modeling strategy in detail below.

**Spectrum preprocessing.** We preprocess each mass spectrum by removing noise peaks and scaling the peak intensities before they are transformed into input embeddings for Casanovo. First, we discard any peaks outside the range 50–2500 $m/z$, as well as any peaks within 2 Da of the observed precursor mass. We then remove any peaks with an intensity value lower than 1% of the most intense peak's intensity, and we retain up to 150 of the most intense peaks in the spectrum. Finally, peaks intensities are square-root transformed and then normalized by dividing by the sum of the square-root intensities.

**Input embeddings.** Each spectrum $S = \{(m_j, I_j)\}_{j=1}^{N}$ is a bag of peaks, where each peak $(m_j, I_j)$ is a 2-tuple representing the $m/z$ value and intensity of the peak. For the task of de novo peptide sequencing, the most important relationships for our model to learn are how the spacing of $m/z$ values between each pair of peaks corresponds to the peptide ions that may have generated them. Secondarily to the spacing of $m/z$ values, the intensity of each peak also contains information about the generating ion; for example, y-ions are generally more intense than b-ions for some fragmentation methods. Given this prior knowledge, we chose embedding methods that would enable Casanovo to learn from the spacing of $m/z$ values and that would emphasize the relative importance of these peak attributes for the de novo sequencing task.

We use a fixed, sinusoidal embedding[21] to project each $m/z$ value to a $d$-dimensional vector, the $m/z$ embedding $f$. Specifically, we create the $m/z$ embedding from an equal number of sine and cosine waveforms spanning the wavelengths from 0.001 to 10,000 $m/z$, where each feature in the embedding $f_i$ is a value from one waveform (Supplementary Fig. S15A). Let $\lambda_{max}$ be the maximum wavelength, $\lambda_{min}$ be the minimum wavelength, $i$ be the index of the feature (zero-based), and $d$ be the number of features. We begin by defining the number of features that are to be represented by sine and cosine waveforms as $d_{sin}$ and $d_{cos}$, respectively:

$$d_{sin} = \left\lceil \frac{d}{2} \right\rceil \tag{1}$$

$$d_{cos} = d - d_{sin} \tag{2}$$

The encoded features are then calculated as:

$$f_i(m_j) = \begin{cases} \sin\left(m_j / \left(\frac{\lambda_{min}}{2\pi} \left(\frac{\lambda_{max}}{\lambda_{min}}\right)^{i/(d_{sin}-1)}\right)\right), & \text{for } i \leq d/2 \\ \cos\left(m_j / \left(\frac{\lambda_{min}}{2\pi} \left(\frac{\lambda_{max}}{\lambda_{min}}\right)^{(i-d_{sin})/(d_{cos}-1)}\right)\right), & \text{for } i > d/2 \end{cases} \tag{3}$$

where $\lambda_{max} = 10,000$ and $\lambda_{min} = 0.001$ in Casanovo.

These input embeddings provide a granular representation of high-precision $m/z$ information and, similar to relative positions in the original transformer model[21], may help the model attend to $m/z$ differences between peaks, which are critical for identification of amino acids in the peptide sequence. In the $m/z$ embeddings, we chose high-frequency waveforms to capture the fine structure present in a mass spectrum, such as that introduced by isotopes and near isobaric species. The waveforms then capture more distant relationships as they progress to lower frequencies; thus, if one were to subtract the $m/z$ of one peak from another, the features that are activated depend on the scale of their relationship. While consecutive b- and y-ions may activate one set of features in $f$, complementary b- and y-ions would likely activate a later set due to their larger $m/z$ difference. Furthermore, the cosine similarity between pairs of $m/z$ embeddings are negatively correlated with their $m/z$ (Supplementary Fig. S15B). We postulate that this property preserves information about $m/z$ distances—which are critical for de novo peptide sequencing—in a manner that is readily accessible to the subsequent transformer layers.

The intensity, which is measured with lower precision than the $m/z$ value, is embedded by projection to $d$ dimensions through a linear layer, after which the $m/z$ and intensity embeddings are summed to produce the input peak embedding. However, in developing

Casanovo, we found that using a sinusoidal encoding for intensity as well as *m/z* performs similarly.

Precursor information, used as input to the decoder, consists of the total mass $m_{prec} \in R$ and charge state $c_{prec} \in \{1, \ldots, 10\}$ associated with the spectrum. We use the same sinusoidal position embedding as peak *m/z*'s for $m_{prec}$; $c_{prec}$ is embedded using an embedding layer, and the embeddings are summed to obtain the input precursor embedding.

**Modeling de novo peptide sequencing as a sequence-to-sequence task.** The transformer architecture in Casanovo follows the standard encoder-decoder design of Vaswani et al.[21] The process begins by embedding the peaks of a mass spectrum to obtain input embeddings $f$, which are then contextualized using the transformer encoder stack. Thus, a full mass spectrum consisting of $k$ peaks is represented as an unordered sequence of peak embeddings $g \in \mathbb{R}^{k \times d}$. The self-attention mechanism of the transformer encoder learns relationships between these peaks and outputs a contextualized embedding of each peak in the mass spectrum $\hat{g} \in \mathbb{R}^{k \times d}$.

The decoding process begins by feeding both the precursor embedding and the contextualized spectrum embedding $\hat{g}$ into the transformer decoder stack. Decoding proceeds in an autoregressive manner; up to a maximum number of iterations, the decoder stack will attempt to predict the next amino acid in the generating peptide from C-terminus to N-terminus. During the decoding phase, a learned representation of the previously predicted amino acid is concatenated to the input into the decoder for each iteration and summed with a sinusoidal positional embedding of its position in the sequence. The output of the decoder are scores $s \in \mathbb{R}^{p \times v}$ representing how confident Casanovo is about each amino acid it has predicted for a peptide sequence of length $p$ and an amino acid vocabulary of size $v$.

**Training and inference strategy.** Taking the previously described embeddings as input, the transformer outputs scores which are treated as a probability distribution over the amino acid vocabulary for the next position in the sequence at each decoding step. The amino acid vocabulary includes 20 canonical amino acids (with cysteine considered to be carbamidomethylated), post-translationally modified versions of three of them (oxidation of methionine and deamidation of asparagine or glutamine), N-terminal modifications (acetylation, carbamylation, loss of ammonia, and the combination of loss of ammonia and carbamylation), plus a special `stop` token to signal the end of decoding, yielding a total of 28 tokens. During training, the decoder is fed the amino acid prefix for the ground truth peptide following the teacher forcing paradigm[44]. Cross-entropy between the model output probabilities and a binary matrix representing the amino acid sequence of the ground truth peptide is minimized as the objective function. During inference, beam search is used to find the highest-scoring predicted peptide sequence, with $k$ a user-specified value for the number of beams. At each prediction step, for every peptide prefix considered, the $k$ top-scoring amino acids are selected, after which the $k$ top-ranked amino acid sequences are used for the subsequent decoding step. Beams are terminated when the `stop` token is predicted, the predicted peptide mass is similar to (given the precursor mass tolerance) or exceeds the precursor mass, or the pre-defined maximum peptide length of $\ell = 100$ amino acids is reached. As final prediction, the top-scoring peptide sequence that fits the precursor mass tolerance (optionally accounting for isotope offsets) is selected. If no peptide prediction fits the precursor mass tolerance, the top-scoring peptide sequence with a non-matching peptide mass is selected.

**Model and training hyperparameters.** We train models with nine layers, embedding size $d = 512$, and eight attention heads, yielding a total of ~47M model parameters. A batch size of 32 spectra and $10^{-5}$

weight decay is used during training, with a peak learning rate of $5 \times 10^{-4}$. The learning rate is linearly increased from zero to its peak value in 100,000 warm-up steps, followed by a cosine shaped decay. The MassIVE-KB model was trained for a single epoch on 30 million PSMs from the MassIVE-KB dataset, which took approximately 8 days on 4 RTX 2080 Ti GPUs, while evaluating the performance on the validation set after every 50,000 iterations. The final model weights were taken from the snapshot with the lowest validation loss. This model was fine-tuned on the non-enzymatic training dataset using a peak learning rate of $5 \times 10^{-5}$. We selected the fine-tuned model with the minimal validation loss, which occurred after 5 epochs. These model hyperparameters—number of layers, embedding size, number of attention heads, and learning rate schedule—are used for all downstream experiments unless otherwise specified.

**Precursor *m/z* filtering.** A critical constraint in de novo peptide sequencing requires the difference between the total mass of the predicted peptide $m_{pred}$ and the observed precursor mass $m_{prec}$ of the spectrum to be smaller than a threshold value $\epsilon$ (specified in ppm) for the predicted sequence to be plausible: $\Delta m_{ppm} = \frac{|m_{prec} - m_{pred}| \times 10^6}{m_{prec}} < \epsilon$. Therefore, in addition to providing precursor information as an input for the model to learn from, we filter out peptide predictions that do not satisfy the above constraint. The threshold value $\epsilon$ is a property of the mass spectrometer that the data is collected with, and hence is known at inference time. Accordingly, we choose $\epsilon$ based on the precursor mass error tolerance used in the database search to obtain ground truth peptide sequences for the test data.

## Datasets

**MassIVE-KB dataset.** A large-scale, heterogeneous dataset derived from the MassIVE knowledge base (MassIVE-KB; v.2018-06-15) was used to develop Casanovo[23]. The MassIVE-KB data set consists of 31 TB of human data from 227 public proteomics datasets, containing over 669 million MS/MS spectra. MassIVE-KB contains a designated subset of 30,506,973 "high quality" PSMs, identified by applying a strict ( ~0%) PSM-level FDR filter and then selecting at most 100 PSMs for each combination of peptide sequence and charge. These 30 million PSMs were randomly split so that the training, validation and test sets are disjoint at the peptide-level and consist of approximately 28 million training PSMs, 1 million validation PSMs, and 1 million test PSMs.

**Nine-species benchmark dataset.** We created a new version of the nine-species benchmark originally described by Tran et al.[8] To do so, we downloaded the RAW files from the same nine PRIDE projects (Supplementary Table S2) and converted them to MGF format using the ThermoRawFileParser v1.3.4. We also downloaded the corresponding nine Uniprot reference proteomes and constructed a Tide index for each one, using Crux version 4.1. For one species (*Vigna mungo*), no reference proteome is available, so we used the proteome of the closely related species *Vigna radiata*. We specified Cys carbamidomethylation as a static modification and allowed for the following variable modifications: Met oxidation, Asn deamidation, Gln deamidation, N-term acetylation, N-term carbamylation, N-term NH₃ loss, and the combination of N-term carbamylation and NH₃ loss by using the tide-index options –mods-spec 1M+15.994915, 1N+0.984016, 1Q+0.984016 –nterm-peptide-mods-spec 1X+42.010565, 1X+43.005814, 1X-17.026549, 1X+25.980265 –max-mods 3. Note that one of the nine experiments (*Mus musculus*) was performed using SILAC labeling, but we searched without SILAC modifications and hence include in the benchmark only PSMs from unlabeled peptides. Each index also contains a shuffled decoy peptide corresponding to each target peptide. Each MGF file was searched against the corresponding index using the precursor window size and fragment bin tolerance specified in the original study (Supplementary Table S2). We used XCorr scoring with Tailor calibration[45], and we allowed for 1 isotope error in the selection of

candidate peptides. All search results were then analyzed jointly per species using the Crux implementation of Percolator, with default parameters. For the benchmark, we retained all PSMs with Percolator q value < 0.01. We identified 13 MGF files with fewer than 100 accepted PSMs, and we eliminated all of these PSMs from the benchmark. We then post-processed the PSMs to eliminate peptides that are shared between species. Among the 229,984 unique peptides, we identified 3797 (1.7%) that occur in more than one species. For each such peptide, we selected one of the associated species at random and then eliminated all PSMs containing that peptide in other species. Note that when identifying shared peptides between species, we considered all modified forms of a given peptide sequence to be the same. Hence, if a given peptide appears in more than one species, then that peptide, including all its modified forms, is randomly assigned to a single species and eliminated from the others. The final benchmark dataset consists of 2.8 million PSMs drawn from 343 RAW files. The revised nine-species benchmark is available on MassIVE at https://doi.org/10.25345/C52V2CK8J.

### Evaluation metrics

We use precision calculated at the amino acid and peptide levels[8,11,13] as a function of coverage over the test set as performance measures to evaluate the quality of a given model's predictions. In each case, for each spectrum we compare the predicted sequence to the ground truth peptide from the database search. Following Tran et al.[8], for the amino acid-level measures we first calculate the number $N_{match}^a$ of matched amino acid predictions, defined as all predicted amino acids which (1) differ by < 0.1 Da in mass from the corresponding ground truth amino acid, and (2) have either a prefix or suffix that differs by no more than 0.5 Da in mass from the corresponding amino acid sequence in the ground truth peptide. We then define amino acid-level precision as $N_{match}^a/N_{pred}^a$, where $N_{pred}^a$ is the number of predicted amino acids. For peptide predictions, a predicted peptide is considered a correct match if all of its amino acids are matched. Among a collection of $N_{orig}^p$ spectra, if our model makes predictions on a subset of $N_{pred}^p$ and correctly predicts $N_{match}^p$ peptides, we define coverage as $N_{pred}^p/N_{orig}^p$ and peptide-level precision as $N_{match}^p/N_{pred}^p$. To plot a precision-coverage curve, we sort predictions by the confidence score provided by the model. Amino acid-level confidence scores are obtained by applying a softmax to the output of the transformer decoder, which is a proxy for the probability of each predicted amino acid to occur in the given position along the peptide sequence. Casanovo reports the mean probability score over all amino acids as a peptide-level confidence score, and reports a modified score at the amino acid level, computed as the mean of the peptide score and the individual amino acid probability score.

### Competing methods

We downloaded DeepNovo weights from https://github.com/nh2tran/DeepNovo/tree/PNAS on Sep 6, 2022. Similar to Casanovo$_{bm}$, DeepNovo and PointNovo were trained in a cross-validated fashion using the original nine-species benchmark, systematically training on eight species and testing on the remaining species. Accordingly, nine different sets of pre-trained DeepNovo weights were available, and the corresponding set of weights were used for testing on each species data set. In the absence of pre-trained PointNovo weights, we cross-validated nine models from scratch by training on eight species and testing on the remaining species. We downloaded the PointNovo code provided by Qiao et al.[17] from https://zenodo.org/records/3960823 on Mar 27, 2023. We downloaded Novor v1.05.0573 from https://github.com/compomics/searchgui/tree/master/resources/Novor on Dec 3, 2022.

### Creating a non-enzymatic dataset

To create a dataset of PSMs that does not exhibit tryptic bias, we selected PSMs with a uniform distribution of amino acids at the C-terminal peptide positions from two datasets: MassIVE-KB[23] and PROSPECT[27]. The MassIVE-KB dataset contains 30 million PSMs and consists entirely of data generated using trypsin; hence, only a small proportion of the MassIVE-KB peptides do not end in K or R, corresponding to those that appear at the C-terminus of the corresponding protein. The PROSPECT dataset is a collection of 61 million PSMs generated from synthetic peptides. We performed three filtering steps on this dataset: (1) removed duplicate peaks with identical $m/z$ values from each spectrum, (2) eliminated spectra with fewer than 20 peaks, and (3) eliminated spectra with Andromeda score less than 70, selecting the highest-scoring peptide for each spectrum. To avoid over-representation of some peptides in this dataset, we randomly selected at most 100 PSMs per unique peptide, similar to the processing that was done during the creation of the MassIVE-KB dataset. This pre-selection step reduced the size of the PROSPECT dataset to 12.6 million PSMs. Finally, to create a non-enzymatic dataset containing 1 million peptides, we selected 50,000 PSMs for each canonical amino acid. These PSMs were selected at random from MassIVE-KB, then supplemented as necessary from PROSPECT to obtain the desired count (Supplementary Table S1). This dataset contained PSMs from 247,859 unique peptides, which were randomly split into training, validation and test sets in the ratio 80/10/10. The non-enzymatic dataset with the training, validation and test splits is available on MassIVE at https://doi.org/10.25345/C5KS6JG0W.

### Immunopeptidome analysis

For the analysis of the MHC class I peptides, we used the Tide search engine and adopted search settings from the original publication[5]. To create the peptide index, we ran tide-index allowing M oxidation or phosphorylation of S/T/Y, with a maximum of one modification per peptide. We set the digestion to be "non-specific-digest," allowed zero missed cleavages, and specified a peptide length range of 8–15 amino acids. Using the canonical human reference proteome downloaded from Uniprot on July 17, 2022, the resulting index contains 286,319,284 peptides and an equal number of shuffled decoy peptides. We searched the data using the tide-index command, specifying a precursor window size of 30 ppm and using Tailor calibration. The resulting sets of PSMs from all five runs were analyzed jointly using Percolator with default settings. All of the above commands were implemented within Crux[46] version 4.1-2fab3c91-2022-11-09. For comparative analysis between Casanovo predictions and database search results, only sequences within a peptide length range of 9–15 amino acids were considered. NetMHCpan-4.1 was used to predict MHC binding affinities for peptide sequences[30]. Binding affinity was predicted for 9-mer amino acid motifs in reference to the HLA-A02:01, HLA-A02:17, HLA-B41:01, HLA-B40:02, HLA-C02:02, and HLA-C17:01 alleles of the MHC molecule.

### Metaproteomics analysis

We analyzed data from six mass spectrometry runs, three replicates each from the Bering Strait (BSt) and the Chukchi Sea (CS)[31], downloaded in mzXML format from https://noble.gs.washington.edu/proj/metapeptide. From the same URL, we also downloaded the two corresponding metapeptide databases, and we downloaded the environmental non-redundant database (env_nr) from NCBI on Nov 12, 2022. We used Tide to build three peptide indices from the two metapeptide databases and from env_nr with default parameters, except we allowed three methionine oxidations per peptide and up to 1 missed cleavage. The resulting indices contained 41,665,963 peptides (CS), 34,116,884 peptides (BSt), and 310,021,565 peptides (env_nr), respectively, as well as

an equal number of shuffled decoy peptides. Each mzXML file was searched against the relevant metapeptide database and against env_nr using the tide-search command, specifying a precursor window of 10 ppm, allowing one isotope error, and using Tailor calibration. Finally, we used Percolator[29] with default parameters to jointly analyze the search results from each set of three runs against the same peptide index. All of the above commands were implemented within Crux[46] version 4.1-2fab3c91-2022-11-09. Metaproteomics spectrum files and peptides detected via database search are available on MassIVE with the dataset identifier MSV000094709 at https://doi.org/10.25345/C5ST7F78Z.

To estimate the error rate for the matching of Casanovo predicted peptides against a protein database, we developed a procedure akin to the target-decoy competition used in false discovery rate estimation for mass spectrometry database search[47]. To do so, we created a decoy protein database by randomly shuffling each protein sequence. We then asked whether each Casanovo prediction appears in the target (i.e., unshuffled) or decoy protein list, marking each predicted peptide as matching to the target, decoy, or neither. Peptides assigned to both were randomly assigned to either the target or decoy. We then segregated the peptides by length, and for each length we sorted the Casanovo predictions by confidence score. At each position $k$ in the ranked list, we estimated the rate of random matching among the target matches as $D_k/T_k$, where $D_k$ (respectively, $T_k$) is the number of decoys (respectively, targets) with rank smaller than $k$. We then selected the largest value of $k$ such that $D_k/T_k < \alpha$, for some specified random matching rate $\alpha$. In this work, we used $\alpha = 0.01$.

We used the online version of Prosit[32] at https://www.proteomicsdb.org/prosit/ to predict peak intensities and retention times using the Prosit_2020_intensity_hcd and Prosit_2019_irt models, respectively. A fixed collision energy of 27 was used for all peptides based on metadata from the spectrum files. Spectrum peaks predicted by Prosit were matched to observed peaks using a 0.05 Da fragment $m/z$ tolerance to calculate the cosine similarity between the predicted and experimental spectra.

### Reporting summary

Further information on research design is available in the Nature Portfolio Reporting Summary linked to this article.

## Data availability

The revised nine-species benchmark is available on MassIVE with the dataset identifier MSV000090982 at https://doi.org/10.25345/C52V2CK8J. The non-enzymatic dataset with training, validation and test splits is available on MassIVE with the dataset identifier MSV000094014 at https://doi.org/10.25345/C5KS6JG0W. Metaproteomics spectrum files and peptides detected via database search are available on MassIVE with the dataset identifier MSV000094709 at https://doi.org/10.25345/C5ST7F78Z. The spectrum identifications for Casanovo and other evaluated tools on the revised nine-species benchmark are available on MassIVE with the dataset identifier MSV000094434 at https://doi.org/10.25345/C5B56DG2T. Similarly, Casanovo-only spectrum identifications for the immunopeptidomics, metaproteomics, and dark matter analyses are available on MassIVE with the dataset identifier MSV000093980 at https://doi.org/10.25345/C5VD6PG45. Source data are provided with this paper.

## Code availability

Casanovo's source code and trained model weights from the MassIVE-KB training and non-enzymatic fine tuning are available under the Apache 2.0 license at https://github.com/Noble-Lab/casanovo[48]. Model weights from the nine-species benchmark training are available at https://doi.org/10.5281/zenodo.10694984. The transformer model was implemented using PyTorch[49] and PyTorch Lightning[50]. Additionally, NumPy[51], Pandas[52], and Scikit-Learn[53] were used for scientific data processing; and spectrum_utils[54], Pyteomics[49], and ppx[55] were used to process MS/MS data. Matplotlib[56] and Seaborn[57] were used for visualization purposes.

Casanovo's source code and trained model weights are available under the Apache 2.0 license at https://github.com/Noble-Lab/casanovo. Casanovo, Casanovo$_{bm}$ and Casanovo$_{ne}$ are all trained using version 4.0.1.

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

## Acknowledgements

This work is in part supported by CCF-2019844 (S.O.) as a part of the NSF Institute for Foundations of Machine Learning (IFML) as well as by National Science Foundation award 2245300 (W.S.N.).

## Author contributions

W.S.N., M.Y., W.B., and W.E.F. conceptualized the work. W.S.N. and S.O. supervised the work. M.Y., W.B., and W.E.F. developed the Casanovo software. M.Y., C.F.M., V.A., W.B., and R.N. carried out the analyses. W.S.N., M.Y., W.B., and W.E.F. wrote the manuscript. All authors reviewed and edited the manuscript.

## Competing interests

The authors declare no conflicts of interest.
