## [Peer Review File · Nature Communications]

Reviewers' Comments:

Reviewer #1:

Remarks to the Author:

The authors present a new version of Casanovo which now incorporates beamsearch and trained on a larger data. This paper is well written and the potential applications presented are interesting. However, my main concern with this work is the lack of technical novelty compared to the previously published work.

Comments/questions

Fig S3 seems to suggest the improvement from beamsearch is quite minimal. It would be nice to see a comparison between different beam sizes.

In NLP, people have observed that there's diminishing returns when increasing the size of the training data. Have you tried training on various amounts of the MassIVE dataset?

With such a large increase in training data, why is the model size kept the same?

It seems a lot of care has been made in order to curate the MassIVE dataset. In my opinion, the effect of the data's cleanliness, distribution, and coverage on the model performance is not well documented in the peptide sequencing literature. Some discussions on the choices made and their effects will be appreciated.

Reviewer #2:

Remarks to the Author:

The manuscript presented Casanovo, a transformer-based DNN model for de novo peptide sequencing from MS/MS spectra. Transformers have shown the power on many deep learning applications involving seq2seq prediction, such as in natural language processing (NLP), audio processing and computer vision. It is thus conceivable to be applied to de novo peptide sequencing even though it is not the first DNN approach to this problem. The manuscript demonstrated the improved performance of Casanovo against DeepNovo and Novar. Most interestingly, it showed it can be applied two scenarios where de novo sequencing are most useful (because no complete reference protein database is available): immunopeptidomics and metaproteomics. The paper is overall well written. However, I have several concerns about the benchmarking results in the paper.

1. The authors should include PointNovo into the benchmarking experiments. Some studies have shown the PointNovo model performs significantly better than the DeepNovo model. Including it in the benchmarking experiments will allow us to assess the advance and advantage of the transformer model. In addition, PointNovo has released the training code; so the authors can re-train the model using the same training dataset for a fair comparison (like the comparison with DeepNovo).

2. Please clarify the purpose of the fine-tuning using non-tryptic peptides. If the goal is to develop a specialized model for immunopeptide sequencing, why not fine-tuning the model using the immunopeptides only? Note that the immunopeptides (bound to MHC) have specific sequence patterns and length constraints different from general non-tryptic peptides, and the model fine-tuned on general non-tryptic peptides may not work optimally on immunopeptides. On the other hand, if the purpose here is to generalize the Casanovo model to non-tryptic peptides (as it is also used for the dark proteome analyses), then the fine-tuned model should also be tested on the tryptic peptides (e.g., the benchmarking datasets from nine species) to see if the performance on tryptic peptides remain the same.

3. The manuscript showed the training on a large spectral library (MassIVE) improve the accuracy of de novo sequencing. Is this also true for DeepNovo and PointNovo? If so, the comparative results should be provided to show this is a general phenomena for DNN-based methods.
4. Are the results presented on the metaproteomic data based on the CasaNovo version fine-tuned on the non-tryptic peptides? If not, are the results also biased to tryptic peptides? Also, it will be useful to report the intersection of the peptides identified by database searching and de novo sequencing, respectively.
5. It will be useful to break down the performance comparison (e.g., in Fig 2) into the spectra of different charges (e.g., 2+ and 3+) for a better understanding of the improvement of the performance.

Reviewer #3:

Remarks to the Author:

This manuscript proposes Casanovo, a new deep learning approach for the task of de novo peptide sequencing from tandem mass (MS/MS) spectra. The proposed model is very closely based on the transformer model from Vaswani et al 2017 (properly acknowledged as reference 21 in the submitted manuscript) but extends it with new embeddings for spectra and amino acid sequences, as well as with a new beam-search approach to define the final output de novo sequence. This is also the full-length journal submission of the approach that was reported in ICML 2022. The overall approach is developed and properly evaluated with sufficient amounts of data and the reported results demonstrate significant improvements over the previous state of the art, with significant improvements in both precision and recall for both full-peptide and amino-acid level sequencing accuracy. While there are some important details missing from the description of the approach and from the presentation of the results (more on this below), it should be noted that it is likely that the significance and novelty of the approach will be further supported by these additional details.

The first major area where the manuscript can be improved is in the description of the embeddings used for both spectra and amino acid sequences, which constitute one of the major novelties of the proposed approach (since most of the internal architecture of transformer model appears to be essentially the same as in Vaswani et al 2017). While the methods section does provide a detailed description of how the spectrum peak masses (or m/z) would be embedded using a sinusoidal encoding, this description could be improved by providing insights into why this should be a good choice of transformation when using this kind of data (MS/MS spectra) for this kind of model. It would also be useful to have a more insightful description of why the linear projection embedding peak intensities into d dimensions results in a vector that makes sense to add to the d-dimension embedding for peak m/z values. In addition, the description of the model could also be improved with a more detailed description of the inputs and outputs of the transformer decoder. All of these embeddings could be made substantially clearer (especially for the mass spectrometry experts who are most likely to benefit from the proposed approach) by providing at least one detailed example for each of these embeddings and accompanying them with intuitive insights supporting the underlying design decisions.

The second area where the manuscript can be improved is in the analysis and presentation of the results.

First, while it is clear that Casanovo outperforms competing approaches, it is likely that its absolute performance (and possibly also relative performance improvements) will vary for peptides of different lengths and precursor charge states. It would thus be important to evaluate the quality of the results for peptides of different lengths, where the boundaries between short, medium and long peptide sequences could either be indicated by the boundaries between modes in the results or by typical boundaries such as short (≤ 9 amino acids), medium (10-18 amino acids) and long

(>18 amino acids).

Second, since the proposed version of Casanovo considers peptides with post-translational modifications, it is important to carefully define the training, test and validation partitions of the datasets in a manner that aggregates all versions of the same peptide sequence (i.e., the unmodified peptide P and all differently-modified versions of the same base sequence P) into only one of these subsets. Since the fragmentation of unmodified and modified peptides is very highly correlated for the set of modifications considered in this manuscript, allowing the differently-modified versions of the same peptide P into more than one of the training, test and validation subsets could allow information to leak into earlier stages of the model and thus bring into question the margin of error for the final results. The authors were careful to consider overlaps between subsets in relation to shared peptides across species so it is possible that this possible issue between unmodified and modified peptides was also already properly addressed but this was not clear from the current version of the manuscript.

Third, the spectrum identifications from Casanovo and from the other methods used for the results reported in the manuscript should be made available online or as supplementary materials to enable a more detailed assessment of how the results compare across the various approaches to result in the figures presented in the manuscript. Also, the comparison of results from different approaches should be supported by Venn diagrams facilitating the visualization of the overlaps between approaches, as well as highlighting results derived uniquely from each approach or disagreeing interpretations for the same spectra. The latter would be especially important to report in the context of providing intuition for the kinds of spectra where Casanovo is most likely to improve on (or disagree with) database searching approaches for the three kinds of data reported in the manuscript: immunopeptidomics, metaproteomics and large scale human proteomics.

In addition to responding to the reviewers' requests, we made several other changes to Casanovo. Since the initial submission of our manuscript, we have received lots of feedback from users. One user identified a minor error in the nine-species benchmark, related to the fact that the Tide search engine was not properly reporting all of the terminal post-translational modifications. We fixed Tide and re-generated the benchmark. A second user identified a discrepancy between the way we described the peak embedding procedure in the paper and what was implemented in the model. We modified the model to match the paper. As a consequence of these changes, the results reported throughout the manuscript are different from those in the original version. Overall, these changes led to relatively small changes in performance, as illustrated in modified figures and text at the end of this document.

Reviewer 1

The authors present a new version of Casanovo which now incorporates beamsearch and trained on a larger data. This paper is well written and the potential applications presented are interesting. However, my main concern with this work is the lack of technical novelty compared to the previously published work.

We thank the reviewer for their positive assessment of our work. Although we agree that the core algorithmic approach is similar to the one described in our previous ICML paper, this work introduces several technical improvements as well as extensions towards novel applications. On the technical front, we have changed the decoding algorithm from a greedy approach to beam search, yielding a boost in performance. More importantly, we have shown that by training on a larger and higher-quality dataset—the MassIVE-KB training data as opposed to the 9-species benchmarking dataset previously—a large boost in performance could be achieved. Our approach is inspired by increasing interest from the machine learning community in understanding how data affects the performance and figuring out ways to use larger datasets with carefully designed quality control to improve performance. Examples include a seminal paper by Zhan et al.¹, and a more recent dataset competition by Gadre et al.². We believe that the demonstrated performance has now reached an inflection point where *de novo* peptide sequencing has truly become a viable option for the unbiased analysis of bottom-up proteomics data. We demonstrate this through strong performance across a wide range of applications, including immunopeptidomics, metaproteomics, and the dark matter analysis. These results are all new compared to the ICML paper. Furthermore, especially considering that the ICML paper is not PubMed indexed, we believe that this work will be of great interest to readers of *Nature Communications*.

Fig S3 seems to suggest the improvement from beamsearch is quite minimal. It would be nice to see a comparison between different beam sizes.

Although the improvement from beam search is indeed rather modest, an important benefit is that this makes it possible to obtain more PSMs that match the precursor mass tolerance, as indicated by the diamond on the precision–coverage plots, thus resulting in more plausible spectrum annotations. Supplementary Figure 3 (Figure 1 in this document) shows a comparison of different beam search sizes, indicating that the peptide identification performance scales with the number of beams used.

In NLP, people have observed that there's diminishing returns when increasing the size of the training data. Have you tried training on various amounts of the MassIVE dataset?

We have now trained Casanovo on progressively larger subsets of the MassIVE-KB data, with the resulting learning curve shown in Supplementary Figure 5 (Figure 2 in this document). This plot is surprising. While it does show improved performance in terms of the training size, the curve relatively quickly flattens off despite an increasing number of spectra or unique peptides. We have described this experiment in the main text as follows:

Figure 1: **Comparison of greedy and beam-search decoding.** (A) The plot shows precision-coverage curves for the Casanovo model, using either greedy decoding or beam-search decoding with different number of beams. The revised 9-species benchmark was used for this analysis. (B) Similar to panel (A), but showing amino acid-level precision and coverage for the same data set.

To better understand why the model trained on MassIVE-KB outperforms the one trained on the 9-species benchmark, we performed two follow-up experiments. First, we trained a series of Casanovo models on randomly sampled nested subsets of MassIVE-KB, ranging from 250,000 spectra to the full dataset of 28 million spectra. Each model was then evaluated with respect to the revised 9-species benchmark. The resulting learning curve (Supplementary Figure 2) shows that the test set performance depends strongly on the size of the training set, though with diminishing returns after a million or so PSMs.

In light of this result, we have also modified our description of our results elsewhere in the main text, as follows:

We hypothesized that Casanovo could achieve even better performance if provided with a larger training set of higher quality PSMs; hence, we turned to the MassIVE-KB spectral library of human MS/MS proteomics data³.

The results from evaluating with respect to this revised benchmark demonstrate the value of training from a much larger collection of higher quality PSMs (Figure 4B). When trained on the MassIVE-KB dataset, the average precision of Casanovo increases from 0.83 to 0.95.

With such a large increase in training data, why is the model size kept the same?

We agree that going from approximately 1 million training spectra in the 9-species benchmark to 30 million spectra in the MassIVE-KB dataset presents opportunities to scale up the model as well. In this case, we decided to stick to our initial model architecture because it presents a good trade-off between high predictive performance, ease of training on commodity hardware (involving being able to load a model on standard consumer GPUs, as were used, and the necessary training runtime), as well as efficiency during inference by users, who might not have advanced GPU setups available. Based on the encouraging results from the

Figure 2: **Casanovo performance on the 9-species benchmark improves with more training data.** Each point corresponds to a Casanovo model trained on one of the nested subsets of MassIVE-KB, ranging from 250,000 spectra to the full dataset of 28 million spectra. Average precision is reported on the revised 9-species benchmark.

learning curve analysis, we reserve exploring larger models for future work while we develop a strategy to make these larger models accessible through an online platform.

It seems a lot of care has been made in order to curate the MassIVE dataset. In my opinion, the effect of the data’s cleanliness, distribution, and coverage on the model performance is not well documented in the peptide sequencing literature. Some discussions on the choices made and their effects will be appreciated.

This is indeed an important aspect that has not been discussed in detail in the literature. Similar to many different machine learning applications, *de novo* sequencing needs high-quality training data to achieve optimal *de novo* peptide sequencing performance. To pinpoint this effect, we performed a head-to-head comparison of two Casanovo models trained on equal-sized data sets derived from MassIVE-KB and the 9-species benchmark Supplementary Figure 6 (Figure 3 in this document):

Second, we directly compared a Casanovo model trained from a downsampled MassIVE-KB dataset to Casanovo_{bm} which averages 9 models cross-validated on the 9-species benchmark, where the training sets contain approximately the same number of peptides (239,697 for MassIVE-KB and 246,713 for Casanovo_{bm}). We then evaluated both models using the revised 9-species benchmark. The results (Supplementary Figure 3) show that the model trained from MassIVE-KB substantially outperforms Casanovo_{bm}, with the average precision increasing from 0.83 to 0.90. Thus, these results suggest that the improved performance of the MassIVE-KB model stems primarily from the improved quality of the data rather than the size of the data set.

We also added the following paragraph to the discussion section emphasizing the importance of high quality data:

Figure 3: **Comparison of Casanovo models trained on the 9-species benchmark and MassIVE-KB.** The figure plots, for each model, precision on the revised 9-species benchmark as a function of coverage. The training sets for Casanovo_{bm} and MassIVE-KB trained Casanovo model contain 246,713 and 239,697 distinct peptides, respectively.

Casanovo’s excellent performance derives from two sources: the availability of a large, high-quality set of training data, and the use of a machine learning architecture that can make use of that data. Our experiments suggest that the carefully curated MassIVE-KB collection provides particularly good training data. This is likely because the dataset was derived from a massive collection of 669 million spectra, in combination with extremely stringent FDR control. In particular, the data were searched at 1% FDR, after which only the top 100 PSMs for each unique precursor were retained, corresponding to 30 million high-quality PSMs (uniformly 0% FDR from the original searches).

Figure 4: **Casanovo outperforms PointNovo, DeepNovo and Novor on a nine-species benchmark.** (A) Casanovo maintains high peptide-level precision (the proportion of correctly predicted peptides) across all values of coverage (the proportion of spectra for which a prediction is made). Each curve is computed by sorting predicted peptides for all nine species according to their peptide-level confidence scores. For Casanovo, all peptides that pass the precursor m/z filter are ranked above peptides that do not pass the filter, and the boundary is indicated by a diamond on each curve. Average precision (AP) corresponds to the area under the precision-coverage curve. (B) Same as panel (A), but using the revised benchmark and including a version of Casanovo trained on MassIVE-KB. (C) Casanovo’s amino acid-level precision is greatly improved by the expanded training data provided by MassIVE-KB. The test set is the revised nine-species benchmark, with PSMs only containing modifications considered by both DeepNovo and Casanovo.

Reviewer 2

The manuscript presented Casanovo, a transformer-based DNN model for de novo peptide sequencing from MS/MS spectra. Transformers have shown the power on many deep learning applications involving seq2seq prediction, such as in natural language processing (NLP), audio processing and computer vision. It is thus conceivable to be applied to de novo peptide sequencing even though it is not the first DNN approach to this problem. The manuscript demonstrated the improved performance of Casanovo against DeepNovo and Novor. Most interestingly, it showed it can be applied two scenarios where de novo sequencing are most useful (because no complete reference protein database is available): immunopeptidomics and metaproteomics. The paper is overall well written. However, I have several concerns about the benchmarking results in the paper.

We thank the reviewer for their suggestions on how to better benchmark Casanovo.

1. The authors should include PointNovo into the benchmarking experiments. Some studies have shown the PointNovo model performs significantly better than the DeepNovo model. Including it in the benchmarking experiments will allow us to assess the advance and advantage of the transformer model. In addition, PointNovo has released the training code; so the authors can re-train the model using the same training dataset for a fair comparison (like the comparison with DeepNovo).

As requested, we have now included PointNovo in the comparisons. These results are now included in Figure 2 (Figure 4 in this document) and show that Casanovo maintains a significant performance advantage compared to PointNovo as well. Following are the relevant updates to the main text and methods sections describing these changes:

To evaluate Casanovo, we first used the nine-species benchmark dataset originally created by Tran *et al.*⁴ to compare the performance of four *de novo* peptide sequencing algorithms: Novor, DeepNovo, PointNovo, and Casanovo. For these comparisons, we used the publicly available, pretrained version of Novor to sequence the MS/MS spectra in the benchmark dataset. DeepNovo,

PointNovo and Casanovo were trained in a cross-validated fashion, systematically training on eight species and testing on the remaining species. For DeepNovo, we used the models trained and provided by Tran *et al.*⁴ for each of the cross-validation splits. For PointNovo, we cross-validated nine models from scratch using the code and settings provided by Qiao *et al.*⁵. This benchmark version of Casanovo, Casanovo_{bm}, employs a simple greedy decoding algorithm, rather than beam-search decoding. The results (Figure 4A) revealed that Casanovo_{bm} substantially improved peptide-level sequencing performance over Novor, DeepNovo and PointNovo, with an average precision of 0.81 for Casanovo_{bm} compared to 0.58, 0.70 and 0.74 for Novor, DeepNovo and PointNovo, respectively.

Similar to Casanovo_{bm}, DeepNovo and PointNovo were trained in a cross-validated fashion using the original nine-species benchmark, systematically training on eight species and testing on the remaining species. Accordingly, nine different sets of pre-trained DeepNovo weights were available, and the corresponding set of weights were used for testing on each species data set. In the absence of pre-trained PointNovo weights, we cross-validated nine models from scratch by training on eight species and testing on the remaining species. We downloaded the PointNovo code provided by Qiao *et al.*⁵ from <https://zenodo.org/records/3960823> on Mar 27, 2023.

2. Please clarify the purpose of the fine-tuning using non-tryptic peptides. If the goal is to develop a specialized model for immunopeptide sequencing, why not fine-tuning the model using the immunopeptides only? Note that the immunopeptides (bound to MHC) have specific sequence patterns and length constraints different from general non-tryptic peptides, and the model fine-tuned on general non-tryptic peptides may not work optimally on immunopeptides. On the other hand, if the purpose here is to generalize the Casanovo model to non-tryptic peptides (as it is also used for the dark proteome analyses), then the fine-tuned model should also be tested on the tryptic peptides (e.g., the benchmarking datasets from nine species) to see if the performance on tryptic peptides remain the same.

The goal of the non-enzymatic fine tuning was to reduce the bias toward tryptic peptides and thereby improve the utility of Casanovo for analyzing data that does not exhibit this bias, such as immunopeptidomics data, but for example also data produced by digestion with alternative enzymes. We agree that it would be possible to go even further to specifically train an immunopeptidomics model, but generating a sufficiently large training dataset for this task is non-trivial. We have now evaluated the non-tryptic model on tryptic data as well, showing a modest decrease in performance, as can be expected (Figure 5 in this document). We have not directly included these results in the manuscript, however, because this small performance drop is expected. Knowing that trypsin was used for digestion is a very strong prior, and hence, the tryptic model should be used for *de novo* sequencing of such data. Only when this is not the case, we recommend using the non-tryptic model.

The non-tryptic model was used for the dark proteome analysis because we were investigating spectra that had already failed to be identified using a standard, tryptic pipeline. Hence, we hypothesized that this set of spectra could be enriched for non-tryptic peptides, leading us to select the non-tryptic model. We have added clarifying text describing these motivations as follows:

Because we were investigating spectra that had already failed to be identified using a standard, tryptic pipeline, we opted to use the non-enzymatic Casanovo model (Casanovo_{ne}) to assign a peptide to each selected MS/MS spectrum, eliminating peptides for which the predicted m/z falls outside the associated mass range.

Figure 5: **Non-enzymatic fine-tuning leads to a modest performance decrease on tryptic test set.** Fine-tuning Casanovo (Casanovo_{ne}) slightly degrades peptide-level precision compared to the standard Casanovo model on the tryptic revised 9-species benchmark.

3. The manuscript showed the training on a large spectral library (MassIVE) improve the accuracy of *de novo* sequencing. Is this also true for DeepNovo and PointNovo? If so, the comparative results should be provided to show this is a general phenomena for DNN-based methods.

Although we expect that alternative methods, such as DeepNovo and PointNovo, could also benefit from being trained on a larger dataset, we did not carry out these trainings. The different *de novo* tools were benchmarked on the nine-species dataset to provide a fair comparison when training them on the same data, demonstrating the improved performance of Casanovo. Repeating the same process on another dataset is non-trivial and is unlikely to result in novel insights. Indeed, it is hard to imagine that training from more (or higher quality, see discussion above) data would not be beneficial for any DNN method.

4. Are the results presented on the metaproteomic data based on the CasaNovo version fine-tuned on the non-tryptic peptides? If not, are the results also biased to tryptic peptides?

Yes, we used the tryptic model. We now explicitly point this out in the main text:

Because these samples were digested using trypsin, we used the standard Casanovo model, trained from the tryptic MassIVE-KB dataset.

Thus, the results are appropriately biased in favor of tryptic peptides.

Also, it will be useful to report the intersection of the peptides identified by database searching and *de novo* sequencing, respectively.

These numbers are now reported, as follows:

When using both metapeptide databases or the non-redundant environment database, Casanovo detects most of the peptides identified by Tide database search and Percolator, where it respectively detects 71% and 75% of Tide identifications on metapeptide and non-redundant databases,

Figure 6: **Breakdown of *de novo* sequencing performance by charge state.** Plots show peptide precision-coverage curves for subsets of the revised nine-species benchmark, grouped by charge state where panels correspond to spectra with (A) 2+ charge, (B) 3+ charge, (C) 4+ or higher charge.

while also detecting a substantial number of additional unique peptides (Supplementary Figure 11).

5. It will be useful to break down the performance comparison (e.g., in Fig 2) into the spectra of different charges (e.g., 2+ and 3+) for a better understanding of the improvement of the performance.

We have now evaluated the Casanovo performance by charge state, as well as by peptide length, as suggested by Reviewer 3. These evaluations are shown in Supplementary Figure 9 (Figure 6 in this document) and Supplementary Figure 10 (Figure 7 in this document), indicating that average precision is lower across all methods for longer peptides and higher charge states, but that Casanovo outperforms its competitors by a larger margin in those settings, especially for peptides longer than 18 amino acids and precursors with 4+ or higher charge state. These details are now reported in the manuscript, as follows:

Second, we bin spectra according to the length of their generating peptides into groups of short (fewer than 13 amino acids), medium (between 13 and 18 amino acids), and long (greater than 18 amino acids) peptides, and compare *de novo* sequencing performance in each group (Supplementary Figure 7). Performance degrades for longer peptides because incorrect amino acid predictions tend to accumulate during decoding, but the observed decrease in average precision for Casanovo is much smaller relative to other methods, highlighting Casanovo’s ability to accurately sequence long peptides as a key contributor to its improved performance.

Figure 7: **Breakdown of *de novo* sequencing performance by peptide length.** Plots show peptide precision-coverage curves for subsets of the revised nine-species benchmark grouped by the length of database search assigned peptides where panels correspond to peptides with (A) fewer than 13 amino acids (B) between 13 and 18 amino acids, (C) greater than 18 amino acids.

Reviewer 3

This manuscript proposes Casanovo, a new deep learning approach for the task of de novo peptide sequencing from tandem mass (MS/MS) spectra. The proposed model is very closely based on the transformer model from Vaswani et al 2017 (properly acknowledged as reference 21 in the submitted manuscript) but extends it with new embeddings for spectra and amino acid sequences, as well as with a new beam-search approach to define the final output de novo sequence. This is also the full-length journal submission of the approach that was reported in ICML 2022. The overall approach is developed and properly evaluated with sufficient amounts of data and the reported results demonstrate significant improvements over the previous state of the art, with significant improvements in both precision and recall for both full-peptide and amino-acid level sequencing accuracy. While there are some important details missing from the description of the approach and from the presentation of the results (more on this below), it should be noted that it is likely that the significance and novelty of the approach will be further supported by these additional details.

We thank the reviewer for the excellent summary of our work and the insightful feedback that has helped make our manuscript stronger.

The first major area where the manuscript can be improved is in the description of the embeddings used for both spectra and amino acid sequences, which constitute one of the major novelties of the proposed approach (since most of the internal architecture of transformer model appears to be essentially the same as in Vaswani et al 2017). While the methods section does provide a detailed description of how the spectrum peak masses (or m/z) would be embedded using a sinusoidal encoding, this description could be improved by providing insights into why this should be a good choice of transformation when using this kind of data (MS/MS spectra) for this kind of model. It would also be useful to have a more insightful description of why the linear projection embedding peak intensities into d dimensions results in a vector that makes sense to add to the d -dimension embedding for peak m/z values. In addition, the description of the model could also be improved with a more detailed description of the inputs and outputs of the transformer decoder. All of these embeddings could be made substantially clearer (especially for the mass spectrometry experts who are most likely to benefit from the proposed approach) by providing at least one detailed example for each of these embeddings and accompanying them with intuitive insights supporting the underlying design decisions.

We agree that these sections did not sufficiently detail our approach to encoding the peaks in a mass spectrum and the peptide sequences. Accordingly, we have revised our methods section to include these details and added Supplementary Figure 15 (Figure 8 in this document) to aid in this description. Furthermore, while preparing this revision a Casanovo user identified a bug in our implementation of the sinusoidal encodings. Although fixing this problem minimally impacted the observed performance of Casanovo, we updated the manuscript to reflect these changes.

Casanovo consists of a transformer encoder and decoder stack as described by Vaswani *et al.*⁶, which are respectively responsible for learning latent representations of the input spectrum peaks and decoding the amino acid sequence of the spectrum’s generating peptide. The encoder takes d -dimensional spectrum peak embeddings as input and outputs d -dimensional latent representation vectors for each peak. Subsequently, the decoder takes as input these representations of prefix amino acids, coupled with a d -dimensional precursor embedding encapsulating precursor m/z and charge information, to predict the next amino acid in the peptide sequence. We discuss different aspects of our modeling strategy in detail below.

Input embeddings

Each spectrum $S = \{(m_j, I_j)\}_{j=1}^N$ is a bag of peaks, where each peak (m_j, I_j) is a 2-tuple representing the m/z value and intensity of the peak. For the task of *de novo* peptide sequencing, the most important relationships for our model to learn are how the spacing of m/z values between each pair of peaks corresponds to the peptide ions that may have generated them.

Secondarily to the spacing of m/z values, the intensity of each peak also contains information about the generating ion; for example, y-ions are generally more intense than b-ions for some fragmentation methods. Given this prior knowledge, we chose embedding methods that would enable Casanovo to learn from the spacing of m/z values and that would emphasize the relative importance of these peak attributes for the *de novo* sequencing task.

We use a fixed, sinusoidal embedding⁶ to project each m/z value to a d -dimensional vector, the m/z embedding f . Specifically, we create the m/z embedding from an equal number of sine and cosine waveforms spanning the wavelengths from 0.001 to 10,000 m/z , where each feature in the embedding f_i is a value from one waveform (Supplementary Figure 8A). Let λ_{\max} be the maximum wavelength, λ_{\min} be the minimum wavelength, i be the index of the feature (zero-based), and d be the number of features. We begin by defining the number of features that are to be represented by sine and cosine waveforms as d_{\sin} and d_{\cos} , respectively:

$$d_{\sin} = \lceil \frac{d}{2} \rceil$$

$$d_{\cos} = d - d_{\sin}$$

The encoded features are then calculated as:

$$f_i(m_j) = \begin{cases} \sin(m_j / (\frac{\lambda_{\min}}{2\pi} (\frac{\lambda_{\max}}{\lambda_{\min}})^{i/(d_{\sin}-1)})), & \text{for } i \leq d/2 \\ \cos(m_j / (\frac{\lambda_{\min}}{2\pi} (\frac{\lambda_{\max}}{\lambda_{\min}})^{(i-d_{\sin})/(d_{\cos}-1)})), & \text{for } i > d/2 \end{cases} \quad (1)$$

where $\lambda_{\max} = 10,000$ and $\lambda_{\min} = 0.001$ in Casanovo.

These input embeddings provide a granular representation of high-precision m/z information and, similar to relative positions in the original transformer model⁶, may help the model attend to m/z differences between peaks, which are critical for identification of amino acids in the peptide sequence. In the m/z embeddings, we chose high-frequency waveforms to capture the fine structure present in a mass spectrum, such as that introduced by isotopes and near isobaric species. The waveforms then capture more distant relationships as they progress to lower frequencies; thus, if one were to subtract the m/z of one peak from another, the features that are activated depend on the scale of their relationship. While consecutive b- and y-ions may activate one set of features in f , complementary b- and y-ions would likely activate a later set due to their larger m/z difference. Furthermore, the cosine similarity between pairs of m/z embeddings are negatively correlated with their m/z (Supplementary Figure 8B). We postulate that this property preserves information about m/z distances—which are critical for *de novo* peptide sequencing—in a manner that is readily accessible to the subsequent transformer layers.

The intensity, which is measured with lower precision than the m/z value, is embedded by projection to d dimensions through a linear layer, after which the m/z and intensity embeddings are summed to produce the input peak embedding. However, in developing Casanovo, we found that using a sinusoidal encoding for intensity as well as m/z performs similarly.

Precursor information, used as input to the decoder, consists of the total mass $m_{\text{prec}} \in R$ and charge state $c_{\text{prec}} \in \{1, \dots, 10\}$ associated with the spectrum. We use the same sinusoidal position embedding as peak m/z 's for m_{prec} : c_{prec} is embedded using an embedding layer, and the embeddings are summed to obtain the input precursor embedding.

Modeling *de novo* peptide sequencing as a sequence-to-sequence task

The transformer architecture in Casanovo follows the standard encoder-decoder design of Vaswani *et al.*⁶. The process begins by embedding the peaks of a mass spectrum to obtain input embeddings f , which are then contextualized using the transformer encoder stack. Thus, a full mass spectrum consisting of k peaks is represented as an unordered sequence of peak embeddings $g \in \mathbb{R}^{k \times d}$. The self-attention mechanism of the transformer encoder learns relationships between these peaks and outputs a contextualized embedding of each peak in the mass spectrum $\hat{g} \in \mathbb{R}^{k \times d}$.

The decoding process begins by feeding both the precursor embedding and the contextualized spectrum embedding \hat{g} into the transformer decoder stack. Decoding proceeds in an autoregressive manner; up to a maximum number of iterations, the decoder stack will attempt to predict

the next amino acid in the generating peptide from c-terminus to n-terminus. During the decoding phase, a learned representation of the previously predicted amino acid is concatenated to the input into the decoder for each iteration and summed with a sinusoidal positional embedding of its position in the sequence. The output of the decoder are scores $s \in \mathbb{R}^{p \times v}$ representing how confident Casanovo is about each amino acid it has predicted for a peptide sequence of length p and an amino acid vocabulary of size v .

The second area where the manuscript can be improved is in the analysis and presentation of the results.

First, while it is clear that Casanovo outperforms competing approaches, it is likely that its absolute performance (and possibly also relative performance improvements) will vary for peptides of different lengths and precursor charge states. It would thus be important to evaluate the quality of the results for peptides of different lengths, where the boundaries between short, medium and long peptide sequences could either be indicated by the boundaries between modes in the results or by typical boundaries such as short (≤ 9 amino acids), medium (10–18 amino acids) and long (> 18 amino acids).

We have now evaluated the Casanovo performance by peptide length, as well as by charge state, which was requested by reviewer 2. These evaluations are shown in Supplementary Figure 10 (Figure 7 in this document) and Supplementary Figure 9 (Figure 6 in this document), indicating that average precision is lower across all methods for longer peptides and higher charge states, but that Casanovo outperforms its competitors by a larger margin in those settings, especially for peptides longer than 18 amino acids and precursors with 4+ or higher charge state. These details are now reported in the manuscript, as follows:

Second, we bin spectra according to the length of their generating peptides into groups of short (fewer than 13 amino acids), medium (between 13 and 18 amino acids), and long (greater than 18 amino acids) peptides, and compare *de novo* sequencing performance in each group (Supplementary Figure 7). Performance degrades for longer peptides because incorrect amino acid predictions tend to accumulate during decoding, but the observed decrease in average precision for Casanovo is much smaller relative to other methods, highlighting Casanovo’s ability to accurately sequence long peptides as a key contributor to its improved performance.

Second, since the proposed version of Casanovo considers peptides with post-translational modifications, it is important to carefully define the training, test and validation partitions of the datasets in a manner that aggregates all versions of the same peptide sequence (i.e., the unmodified peptide P and all differently-modified versions of the same base sequence P) into only one of these subsets. Since the fragmentation of unmodified and modified peptides is very highly correlated for the set of modifications considered in this manuscript, allowing the differently-modified versions of the same peptide P into more than one of the training, test and validation subsets could allow information to leak into earlier stages of the model and thus bring into question the margin of error for the final results. The authors were careful to consider overlaps between subsets in relation to shared peptides across species so it is possible that this possible issue between unmodified and modified peptides was also already properly addressed but this was not clear from the current version of the manuscript.

Thank you for this insightful comment. We had previously not ensured that peptidofoms with the same base sequences should always be assigned to the same data split. We have now re-done the train, validation, and test splits for the nine-species benchmark, MassIVE-KB, and the non-enzymatic data to enforce this constraint. We retrained all models discussed in the manuscript and have updated all results accordingly. These results indicate that, fortunately, our previous results did not significantly suffer from overfitting, as the newly obtained results are extremely similar. The updated description of the data splitting procedures in the methods is as follows:

Figure 8: **Sinusoidal encodings represent m/z distance between peaks in a mass spectrum.** (A) The m/z value of each peak is encoded from a progression of sinusoids defined by a minimum and maximum wavelength. In this example, a 6-dimensional embedding ($d = 6$) of m/z 1.0 and m/z 3 is created from sinusoids ranging from a wavelength of m/z 1 ($\lambda_{\min} = 1$) to 10 ($\lambda_{\max} = 10$) to demonstrate how the encoding is performed. (B) The Casanovo sinusoidal embeddings are 512-dimensional ($d = 512$) and created from sinusoids ranging from m/z 0.0001 ($\lambda_{\min} = 0.0001$) to 10,000 ($\lambda_{\max} = 10,000$). The utility of these embeddings lies in their preservation of m/z distance in their embedded space. Here, we sample 10,000 pairs of m/z values between m/z 0 and 2000. The cosine similarity between these embeddings is negatively correlated with the original distance between m/z values.

Note that when identifying shared peptides between species, we considered all modified forms of a given peptide sequence to be the same. Hence, if a given peptide appears in more than one species, then that peptide, including all its modified forms, is randomly assigned to a single species and eliminated from the others.

Third, the spectrum identifications from Casanovo and from the other methods used for the results reported in the manuscript should be made available online or as supplementary materials to enable a more detailed assessment of how the results compare across the various approaches to result in the figures presented in the manuscript.

We have now deposited all spectrum identifications to MassIVE, with dataset MSV000093979 containing the results for all evaluated tools on the nine-species benchmark data, and dataset MSV000093980 containing the Casanovo-only results for the immunopeptidomics, metaproteomics, and dark matter analyses. This information has similarly been listed in the manuscript in the data availability statement.

Also, the comparison of results from different approaches should be supported by Venn diagrams facilitating the visualization of the overlaps between approaches, as well as highlighting results derived uniquely from each approach or disagreeing interpretations for the same spectra. The latter would be especially important to report in the context of providing intuition for the kinds of spectra where Casanovo is most likely to improve on (or disagree with) database searching approaches for the three kinds of data reported in the manuscript: immunopeptidomics, metaproteomics and large scale human proteomics.

We have now included upset plots to compare the overlap (i) in terms of spectrum identifications for all tools on the nine-species benchmark dataset (Supplementary Figure 8; Figure 9 in this document) and (ii) between Casanovo and Tide database search for the immunopeptidomics (Supplementary Figure 11; Figure 10 in this document) and metaproteomics use cases (Supplementary Figure 12; Figure 11 in this document). These results show that correct Casanovo PSMs on the nine-species benchmark dataset include almost all correct identifications of the competing *de novo* sequencing methods as well as approximately 50% more correct PSMs that are unique to Casanovo. Similarly, in the immunopeptidomics and metaproteomics analyses, Casanovo detects most of the peptides identified by Tide database search and Percolator, while also detecting a substantial number of additional peptides. Note that no overlap is shown for the dark proteome analysis, because this set is empty by definition: we analyzed spectra that remained unannotated after sequence database searching.

References

1. Zhan, F. *et al.* Multimodal image synthesis and editing: A survey and taxonomy. *IEEE Transactions on Pattern Analysis and Machine Intelligence* (2023).
2. Gadre, S. *et al.* DataComp: In search of the next generation of multimodal datasets. *arXiv preprint arXiv:2304.14108* (2023).
3. Wang, M. *et al.* Assembling the Community-Scale Discoverable Human Proteome. *Cell Systems* **7**, 412–421.e5 (4 2018).

Figure 9: **Casanovo expands the number of correct PSMs identified by its competitors on the nine-species benchmark.** The plot shows the overlap in peptide predictions between Casanovo and three competing *de novo* sequencing methods for the nine-species benchmark dataset. For each subset of PSMs, black circles denote whether the corresponding method is correct. Horizontal bars indicate the total number of correct PSMs for each method.

Figure 10: **Casanovo identifies a greater number of immunopeptides than Tide database search.** The plot shows the overlap between unique peptides assigned by Casanovo that match to the human proteome and by Tide at 1% FDR for the immunopeptidomics dataset.

Figure 11: **Casanovo detects a substantial number of additional peptides compared to Tide database search in metaproteomics samples.** (A) The plot shows the overlap between unique peptides assigned by Casanovo at 1% error rate and Tide at 1% FDR when respective metapeptide databases are used for the Bering Sea and the Chukchi Sea datasets. (B) Similar to panel (A), but the non-redundant environment database is used for searching and error rate control.

- Tran, N. H., Zhang, X., Xin, L., Shan, B. & Li, M. De novo peptide sequencing by deep learning. *Proceedings of the National Academy of Sciences of the United States of America* **31**, 8247–8252 (2017).
- Qiao, R. *et al.* Computationally instrument-resolution-independent de novo peptide sequencing for high-resolution devices. *Nature Machine Intelligence* **3**, 420–425 (2021).
- Vaswani, A. *et al.* Attention Is All You Need. en. *Advances in Neural Information Processing Systems* **30** (2017).

Reviewers' Comments:

Reviewer #1:

Remarks to the Author:

The paper presents Casanovo a transformer-based de novo sequencing model. This work is an extension to a previously published conference paper. The main contribution of this work is to show the possibility of training transformer-based sequencing model in a large well-curated data. The paper provides extensive experiments to help validates the performance and limitations of the model. It also includes experiments that showcases the potential of de novo sequencing in many applications.

I have read the rebuttal to my and other reviewers' comments. It has properly addressed my concerns. I think this work should be published as it will be very beneficial to the readers of Nature Communications.

Some final comments: I agree that experimenting on model scale (or a Chinchilla-like scaling law) is beyond the scope of this paper, but some mention of this might be included in the discussion or limitation section.

Reviewer #2:

Remarks to the Author:

I appreciate the authors extensive efforts to improve their manuscript. The revision has addressed most of my concerns, and has provided justifiable explanation for those they chose not to pursue. I have no further concerns. But I would suggest the authors to include the references of two recent publications on deep learning-based de novo sequencing algorithms. As they are published after the original submission of this manuscript, I do not suggest further benchmarking comparison against them. However, I think it is useful to reference these papers.

[1] Liu, K., Ye, Y., Li, S. and Tang, H., 2023. Accurate de novo peptide sequencing using fully convolutional neural networks. Nature Communications, 14(1), p.7974.

[2] Yang, T., Ling, T., Sun, B., Liang, Z., Xu, F., Huang, X., Xie, L., He, Y., Li, L., He, F. and Wang, Y., 2024. Introducing n-HelixNovo for practical large-scale de novo peptide sequencing. Briefings in Bioinformatics, 25(2), p.bbbae021.

Reviewer #3:

Remarks to the Author:

The revised version of the manuscript provided clear explanations and additional details for parts of the text that were not as clear before and thus clarified the related important questions about the proposed approach. However, the revised submission only partially addressed important concerns related to missing supporting information - detailed inspection of the data supporting the manuscript has revealed additional concerns whose clarification would require the authors to submit `_all_` supporting information, as had been requested in the previous round of review.

Regarding the metaproteomics datasets, the Casanovo results submitted during the manuscript revision (MSV000093979 and MSV000093980) contain the Casanovo results but are missing all the spectrum files and all the Tide results that were used to evaluate the performance of Casanovo in the manuscript (there are also no protein assignments for Casanovo results so it's not clear which peptides matched to the database and which did not). This is worrisome since the only results the authors provide for database search of the same spectra are for Comet (not Tide as mentioned in the manuscript) and a direct comparison of results between Casanovo and Comet shows many disagreements (agreement rates drop to 50% after just a few top hundred spectrum identifications). While this could be related to issues in the old Comet searches that were later revised in the reported Tide searches, it is not possible to verify this unless the authors properly submit all spectra and search results (Tide included) that were used for the reported comparisons (and this should be on a public repository, not on the authors' web site). Only then it would be possible to establish that the reported Casanovo gains in identifications were not improperly affected by any issues in the database search results used as the baseline for the comparisons.

The supporting information for the 9 species dataset unfortunately appears to be in an even less

usable format than for the metaproteomics dataset – the Casanovo results are only provided for spectra aggregated in single MGF file per species, so there is no way to link the Casanovo results back to the raw files in the original datasets. Without this information for all datasets it is not possible to use the supporting data to confirm the reported results or to inspect the validity of potentially-surprising Casanovo-only results.

Minor comments:

- Figure S8 is not cited in the manuscript.

- The manuscript mentioned that provided results for Casanovo and Casanovo_ne used version 3.2.0. However, this is not consistent with the information in the provided mzTab file in the dataset MSV000093979 and MSV000093980, which indicates that these results were run using version 4.0.1?

- Spectrum preprocessing was not explained in the manuscript. The source code in the GitHub repository indicates that filtering and transformations were applied to the input spectra: 1) Limiting m/z ranges, 2) Removing the precursor peak, 3) Filtering low intensity peaks, 4) Scaling the spectrum intensity using square root and 5) Normalising the spectrum intensities to a unit vector. Not mentioning these in the manuscript may convey the incorrect impression that the model can work with unprocessed spectra.

Reviewer 1

The paper presents Casanovo a transformer-based de novo sequencing model. This work is an extension to a previously published conference paper. The main contribution of this work is to show the possibility of training transformer-based sequencing model in a large well-curated data. The paper provides extensive experiments to help validates the performance and limitations of the model. It also includes experiments that showcases the potential of de novo sequencing in many applications.

I have read the rebuttal to my and other reviewers' comments. It has properly addressed my concerns. I think this work should be published as it will be very beneficial to the readers of Nature Communications.

We thank the reviewer for this summary and positive review.

Some final comments: I agree that experimenting on model scale (or a Chinchilla-like scaling law) is beyond the scope of this paper, but some mention of this might be included in the discussion or limitation section.

We have added the following sentence to the Discussion section:

One important open question, which we leave for future work, is how the number of model parameters impacts *de novo* sequencing performance.

(Remarks on code availability):

I have looked over the documents and the code of the main model. It's well documented and demonstrated the engineering effort put into it.

Reviewer 2

I appreciate the authors extensive efforts to improve their manuscript. The revision has addressed most of my concerns, and has provided justifiable explanation for those they chose not to pursue. I have no further concerns. But I would suggest the authors to include the references of two recent publications on deep learning-based *de novo* sequencing algorithms. As they are published after the original submission of this manuscript, I do not suggest further benchmarking comparison against them. However, I think it is useful to reference these papers.

[1] Liu, K., Ye, Y., Li, S. and Tang, H., 2023. Accurate *de novo* peptide sequencing using fully convolutional neural networks. *Nature Communications*, 14(1), p.7974.

[2] Yang, T., Ling, T., Sun, B., Liang, Z., Xu, F., Huang, X., Xie, L., He, Y., Li, L., He, F. and Wang, Y., 2024. Introducing π -HelixNovo for practical large-scale *de novo* peptide sequencing. *Briefings in Bioinformatics*, 25(2), p.bbbae021.

We agree with the reviewer, and we have added the following paragraph to the Discussion section of our paper, citing these and several other deep learning methods that have been published recently:

The potential for deep learning methods to improve our ability to perform *de novo* sequencing has now been widely recognized. While this paper was under review, at least six additional deep learning *de novo* sequencing methods have been published, including GraphNovo [1], PepNet [2], Denovo-GCN [3], Spectralis [4], π -HelixNovo [5], and NovoB [6]. Clearly, the field would benefit from an exhaustive and rigorous benchmark comparison of this growing field of tools.

Reviewer 3

The revised version of the manuscript provided clear explanations and additional details for parts of the text that were not as clear before and thus clarified the related important questions about the proposed approach. However, the revised submission only partially addressed important concerns related to missing supporting information - detailed inspection of the data supporting the manuscript has revealed additional concerns whose clarification would require the authors to submit all supporting information, as had been requested in the previous round of review.

We appreciate the reviewer's attention to these important supplementary materials and their interest in verifying our results. We now provide additional supplementary information, as outlined below.

Regarding the metaproteomics datasets, the Casanovo results submitted during the manuscript revision (MSV000093979 and MSV000093980) contain the Casanovo results but are missing all the spectrum files and all the Tide results that were used to evaluate the performance of Casanovo in the manuscript (there are also no protein assignments for Casanovo results so it's not clear which peptides matched to the database and which did not). This is worrisome since the only results the authors provide for database search of the same spectra are for Comet (not Tide as mentioned in the manuscript) and a direct comparison of results between Casanovo and Comet shows many disagreements (agreement rates drop to 50% after just a few top hundred spectrum identifications). While this could be related to issues in the old Comet searches that were later revised in the reported Tide searches, it is not possible to verify this unless the authors properly submit all spectra and search results (Tide included) that were used for the reported comparisons (and this should be on a public repository, not on the authors' web site). Only then it would be possible to establish that the reported Casanovo gains in identifications were not improperly affected by any issues in the database search results used as the baseline for the comparisons.

The Comet results referred to above are from the original metaproteomics paper where this data was described, and those results were not used in our analyses. To address the reviewer's concerns, we have now submitted to MassIVE the metaproteomics dataset and associated peptide detections from Tide and Percolator, and we have added the following sentence to the manuscript:

Metaproteomics spectrum files and peptides detected via database search are available on MassIVE at <https://doi.org/doi:10.25345/C5SB3X91X>.

The supporting information for the 9 species dataset unfortunately appears to be in an even less usable format than for the metaproteomics dataset – the Casanovo results are only provided for spectra aggregated in single MGF file per species, so there is no way to link the Casanovo results back to the raw files in the original datasets. Without this information for all datasets it is not possible to use the supporting data to confirm the reported results or to inspect the validity of potentially-surprising Casanovo-only results.

We apologize for this oversight, and we agree with the reviewer that providing the information about the mapping from spectra to MGF filenames is important. The Casanovo results have accordingly been updated with the run names and scan numbers in this MassIVE submission: <https://doi.org/doi:10.25345/C5B56DG2T>.

Minor comments:

Figure S8 is not cited in the manuscript.

We thank the reviewer for pointing out the missing reference to Supplementary Figure S8 in the manuscript. We have added the following description to our Results section:

Additionally, an analysis of spectrum identifications for all *de novo* sequencing tools on the nine-species benchmark dataset shows that correct Casanovo PSMs include almost all correct

identifications of the competing *de novo* sequencing methods, as well as approximately 50% more correct PSMs that are unique to Casanovo (Supplementary Figure ??).

The manuscript mentioned that provided results for Casanovo and Casanovo_{ne} used version 3.2.0. However, this is not consistent with the information in the provided mzTab file in the dataset MSV000093979 and MSV000093980, which indicates that these results were run using version 4.0.1?

We apologize for this oversight. In preparing our revised manuscript, Casanovo version 4.0.1 was used to train the models and generate all results. We have fixed this in the manuscript.

Casanovo, Casanovo_{bm} and Casanovo_{ne} are all trained using version 4.0.1.

Spectrum preprocessing was not explained in the manuscript. The source code in the GitHub repository indicates that filtering and transformations were applied to the input spectra: 1) Limiting m/z ranges, 2) Removing the precursor peak, 3) Filtering low intensity peaks, 4) Scaling the spectrum intensity using square root and 5) Normalising the spectrum intensities to a unit vector. Not mentioning these in the manuscript may convey the incorrect impression that the model can work with unprocessed spectra.

We thank the reviewer for pointing out that a description of spectrum preprocessing steps was missing. We have added the following paragraph to the Methods section of our manuscript:

We preprocess each mass spectrum by removing noise peaks and scaling the peak intensities before they are transformed into input embeddings for Casanovo. First, we discard any peaks outside the range 50–2500 *m/z*, as well as any peaks within 2 Da of the observed precursor mass. We then remove any peaks with an intensity value lower than 0.01% of the most intense peak’s intensity, and we retain up to 150 of the most intense peaks in the spectrum. Finally, peaks intensities are square-root transformed and then normalized by dividing by the sum of the square-root intensities.

(Remarks on code availability):

Unfortunately the results in the paper are not reproducible because there is missing supporting information, as has been requested from the authors in the main text of the review.

Please see our response to this concern above.

References

1. Mao, Z., Zhang, R., Xin, L. & Li, M. Mitigating the missing fragmentation problem in *de novo* peptide sequencing with a two stage graph-based deep learning model. *Nature Machine Intelligence* **5** (2023).
2. Liu, K., Ye, Y., Li, S. & Tang, H. Accurate *de novo* peptide sequencing using fully convolutional neural networks. *Nature Communications* **14**, 7974 (2023).
3. Wu, R., Zhang, X., Wang, R. & Wang, H. Denovo-GCN: De Novo Peptide Sequencing by Graph Convolutional Neural Networks. *Applied Sciences* **13** (2023).
4. Klaproth-Andrade, D. *et al.* Deep learning-driven fragment ion series classification enables highly precise and sensitive *de novo* peptide sequencing. *Nature Communications* **15**, 151 (2024).
5. Yang, T. *et al.* Introducing π -HelixNovo for practical large-scale *de novo* peptide sequencing. *Briefings in Bioinformatics* **25**, bbae021 (2024).
6. Lee, S. & Kim, H. Bidirectional *de novo* peptide sequencing using a transformer model. *PLOS Computational Biology* **20**, e1011892 (2024).

Reviewers' Comments:

Reviewer #3:

Remarks to the Author:

The authors have satisfactorily addressed almost all concerns indicated in the previous round of reviews. Even though some important supplementary information is still missing, the overall assessment is that (a) the models are sound and the overall approach is reasonable and well described, (b) the results presented in the manuscript are plausible, (c) there is no reason to doubt the integrity of the authors in having intended to assess the performance of Casanovo to the best of their ability.

That said, the one pending concern (which could reasonably be considered minor enough to not require an additional revision) that was not properly addressed in the current revision was the request to provide enough supplementary data for the metaproteomics results to be able "to establish that the reported Casanovo gains in identifications were not improperly affected by any issues in the database search results used as the baseline for the comparisons". To be able to do this, it is necessary to provide the FDR-filtered spectrum-level identifications obtained with both Tide and Casanovo to make it possible (a) to evaluate which spectra were not identified by Tide but were identified by Casanovo even though the correct peptide was in the database and (b) to evaluate the level of agreement/disagreement between Tide and Casanovo identifications for the same spectra. Unfortunately the Tide results that were made available were reported only at the peptide level with no information for the spectrum-level identifications so it does not support this assessment of the possible underlying reasons for the reported increase in performance.